# Unlocking Deterministic Robustness Certification on ImageNet

**Kai Hu**
Carnegie Mellon University
Pittsburgh, PA 15213
kaihu@andrew.cmu.edu

**Andy Zou**
Carnegie Mellon University
Pittsburgh, PA 15213
andyzou@cmu.edu

**Zifan Wang**
Center for AI Safety
San Francisco, CA 94111
zifan@safe.ai

**Klas Leino**
Carnegie Mellon University
Pittsburgh, PA 15213
kleino@cs.cmu.edu

**Matt Fredrikson**
Carnegie Mellon University
Pittsburgh, PA 15213
mfredrik@cs.cmu.edu

## Abstract

Despite the promise of Lipschitz-based methods for provably-robust deep learning with deterministic guarantees, current state-of-the-art results are limited to feed-forward Convolutional Networks (ConvNets) on low-dimensional data, such as CIFAR-10. This paper investigates strategies for expanding certifiably robust training to larger, deeper models. A key challenge in certifying deep networks is efficient calculation of the Lipschitz bound for residual blocks found in ResNet and ViT architectures. We show that fast ways of bounding the Lipschitz constant for conventional ResNets are loose, and show how to address this by designing a new residual block, leading to the *Linear ResNet* (LiResNet) architecture. We then introduce *Efficient Margin MAximization* (EMMA), a loss function that stabilizes robust training by penalizing worst-case adversarial examples from multiple classes simultaneously. Together, these contributions yield new *state-of-the-art* robust accuracy on CIFAR-10/100 and Tiny-ImageNet under $\ell_2$ perturbations. Moreover, for the first time, we are able to scale up fast deterministic robustness guarantees to ImageNet, demonstrating that this approach to robust learning can be applied to real-world applications. Our code is publicly available on GitHub.[1]

## 1 Introduction

Deep neural networks have been shown to be vulnerable to well-crafted tiny perturbations, also known as adversarial examples [19, 45]. Methods for achieving provably-robust inference—that is, predictions that are guaranteed to be consistent within a norm-bounded $\epsilon$-ball around an input—are desirable in adversarial settings, as they offer guarantees that hold up against arbitrary perturbations. Thus, The focus of this paper is to train a network that can *certify* its predictions within an $\epsilon$-ball.

Over the last few years, a wide body of literature addressing robustness certification has emerged [6, 7, 18, 24, 26, 29, 32, 33, 43, 46, 48, 52]. To date, the methods that achieve the best certified performance are derived from *randomized smoothing* (RS) [6]; however, this approach has two main drawbacks. First, it provides only a *probabilistic* guarantee, which may generate a false positive claim around 0.1% of the time [6]; by contrast, *deterministic* certification may be preferred in safety-critical applications, e.g., malware detection and autonomous driving. Additionally, RS requires significant

---

[1]Code available at https://github.com/hukkai/liresnet.

computational overhead for both evaluation and certification—this limitation is severe enough that RS methods are typically evaluated on a 1% subset of the ImageNet validation set for timing concerns.

Because of the highly non-linear boundaries learned by a neural network, deterministically certifying the robustness of its predictions usually requires specialized training procedures that regularize the network for efficient certification, as post-hoc certification is either too expensive [27, 44] or too imprecise [18], particularly as the scale of the model being certified grows.

The most promising such approaches—in terms of both certified accuracy and efficiency— perform certification using Lipschitz bounds. For this to work, the learning procedure must impose Lipschitz constraints during training, either through regularization [32] or orthogonalization [1, 42, 48]. While Lipschitz-based certification is efficient enough to perform robustness certification at scale (e.g., on ImageNet) in principle, it imposes strict regularization that makes

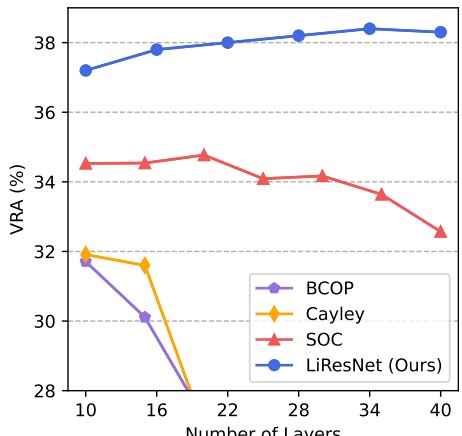

Figure 1: Plot of VRA—the % of points where the model is both correct and certifiably robust— against the depth of the model. We compare our proposed architecture with BCOP [33], SOC [43] and Cayley layers [48] on CIFAR-100, finding our architecture scales more favorably to deeper nets. See Appendix E for plot implementation details.

training large models difficult, especially as it is challenging to maintain tight Lipschitz bounds for very deep models. As a result, state-of-the-art (deterministically) certified robustness is currently still achieved using relatively small feed-forward Convolutional Networks (ConvNets). At the same time, recent work has suggested that robust learning requires additional network capacity [4, 30], leaving the plausibility of deterministic certification for realistic applications using small ConvNet architectures in question.

The goal of this work is to scale up certifiable robustness training from ConvNets to larger, deeper architectures, with the aim of achieving higher *Verifiable Robust Accuracy* (VRA)—the percentage of points on which the model is both correct and certifiably robust. By realizing gains from more powerful architectures, we show it is possible to obtain non-trivial certified robustness on larger datasets like ImageNet, which, to our knowledge, has not yet been achieved by deterministic methods. Our results stem chiefly from two key innovations on *GloRo Nets* [32], a leading certified training approach.[2] First, we find that the residual branches used in conventional ResNet and ViT architectures might not be a good fit for Lipschitz-based certification; instead, we find the key ingredient is to use a *linear* residual path (Figure 2a) forming what we refer to as a *LiResNet* block. The motivation for this architecture development is covered in depth in Section 3.

Second, we find that the typical loss function used by GloRo Nets in the literature may be suboptimal at pushing the decision boundary far away from the input in learning settings with large numbers of classes. Specifically, the standard GloRo loss penalizes possible adversarial examples from a *single* class at once (corresponding to the *closest* adversarial example), while we find a robust model can be more efficiently obtained by regularizing against adversarial examples from *all* possible classes at once, particularly when there are many classes. We thus propose *Efficient Margin MAximization* (EMMA) loss for GloRo Nets, which simultaneously handles possible adversarial examples from any class. More details on the construction and motivation behind EMMA loss are provided in Section 4.

Using the LiResNet architecture and EMMA loss, we are able to (1) scale up deterministic robustness guarantees to ImageNet for the first time, and (2) substantially improve VRA on the benchmark datasets used in prior work. In particular, as examplified in Figure 1, in contrast to the existing architectures that achieve no gains in VRA when going deeper, LiResNet demonstrates an effective use of the increasing capacity to learn more robust functions.

---

[2]We choose GloRo Nets in particular because they involve the least run-time and memory overhead compared to other leading certification methods, e.g., orthogonalization.

To summarize our contributions: (1) we introduce LiResNet, a ResNet architecture using a *linear residual branch* that better fits the Lipschitz-based certification approach for training very deep networks; and (2) we propose EMMA loss to improve the training dynamics of robust learning with GloRo Nets. As a result, we achieve the new state-of-the-art VRAs on CIFAR-10 (70.1%), CIFAR-100 (41.5%), Tiny-ImageNet (33.6%) and ImageNet (35.0%) for $\ell_2$-norm-bounded perturbations with a radius $\epsilon = {}^{36}/{}_{255}$. More importantly, to the best of our knowledge, for the first time we are able to scale up deterministic robustness guarantee to ImageNet, demonstrating the promise, facilitated by our architecture, of obtaining certifiably robust models in real-world applications.

## 2    Background

We are interested in certifying the predictions of a network $F(x) = \arg\max_j f_j(x)$ that classifies an input $x \in \mathbb{R}^d$ into $m$ classes. We use the uppercase $F(x) : \mathbb{R}^d \to [m]$ to denote the predicted class and the lowercase $f(x) : \mathbb{R}^d \to \mathbb{R}^m$ for the logits. A certifier checks for whether $\epsilon$-*local robustness* (Definition 1) holds for an input $x$.

**Definition 1** ($\epsilon$-Local Robustness). *A network $F(x)$ is $\epsilon$-locally robust at an input $x$ w.r.t norm, $||\cdot||$, if $\forall x' \in \mathbb{R}^d . ||x' - x|| \leq \epsilon \implies F(x') = F(x)$.*

Existing certifiers are either *probabilistic* (i.e., guaranteeing robustness with bounded uncertainty) or *deterministic* (i.e., returning a certificate when a point is guaranteed to be robust). We focus on the latter in this paper and consider the $\ell_2$ norm if not otherwise noted.

**Lipschitz-based Certification.**    Certifying the robustness of a prediction can be achieved by checking if the margin between the logit of the predicted class and the others is large enough that no other class will surpass the predicted class on any neighboring point in the $\epsilon$-ball. To this end, many prior works rely on calculating the Lipschitz Constant $K$ (Definition 2) of the model to bound the requisite margin.

**Definition 2** ($K$-Lipschitz Function). *A function $h : \mathbb{R}^d \to \mathbb{R}^m$ is $K$-Lipschitz w.r.t $S \subseteq \mathbb{R}^d$ and norm, $||\cdot||$, if $\forall x, x' \in S . ||h(x) - h(x')|| \leq K||x - x'||$.*

Here, $K$ is the maximum change of a function's output for changing the input in $S$. Notice that it is sufficient to use a *local* Lipschitz Constant $K_{\text{local}}$ of the model, i.e., $S = \{x' : ||x' - x|| \leq \epsilon\}$ in Definition 2, to certify robustness [55]. However, a local bound is often computationally expensive and may need a bounded activation in training [24]. Alternatively, one can compute a *global* Lipschitz Constant $K_{\text{global}}$, i.e. $S = \mathbb{R}^d$ in Definition 2, and leverage the relation $K_{\text{global}} \geq K_{\text{local}}$ to certify any input at any radius $\epsilon$. A global bound is more efficient at test time for certification because it only needs to be computed once and can be used for any input at any $\epsilon$.[3] However, an arbitrary global bound can be vacuously large and thus not useful for positive certification.

**GloRo Nets.**    Among leading approaches that tighten the global Lipschitz constant during training for the purpose of certification [32, 42, 43, 48], GloRo, proposed by Leino et al. [32], is unique in that it naturally incorporates Lipschitz regularization into its loss rather than imposing direct constraints on the Lipschitz constant of each layer. As a result, it is more resource-efficient, and thus has the best potential to scale up to large-scale networks with high-dimensional inputs. GloRo computes the global Lipschitz constant, $K_{ji}$ of the logit margin between the predicted class, $j$, and *every other class*, $i$, by taking the product of the Lipschitz constant of each constituent layer. Next, suppose the Lipschitz constant of the model $F$ until the penultimate layer is $K_{:-1}$, and use $w_i$ as the $i^{\text{th}}$ column of the top layer's weight matrix; then GloRo computes a logit score $f_\perp$ of an artificial class, which we will refer to as $\perp$.

$$f_\perp(x) = \max_{i \neq j}\{f_i(x) + \epsilon K_{ji}\} \text{ where } K_{ji} = ||w_j - w_i|| \cdot K_{:-1}. \tag{1}$$

Then, the so-called *GloRo Net* outputs $F(x) = j$ when $f_j(x) \geq f_\perp(x)$, or $F(x) = \perp$ otherwise. Thus, by design, whenever the GloRo Net does not output $\perp$, its prediction is guaranteed to be robust.

---

[3]Furthermore, when the network is *trained for certification with the global bound*, the global bound has the same certification potential as the local bound [32].

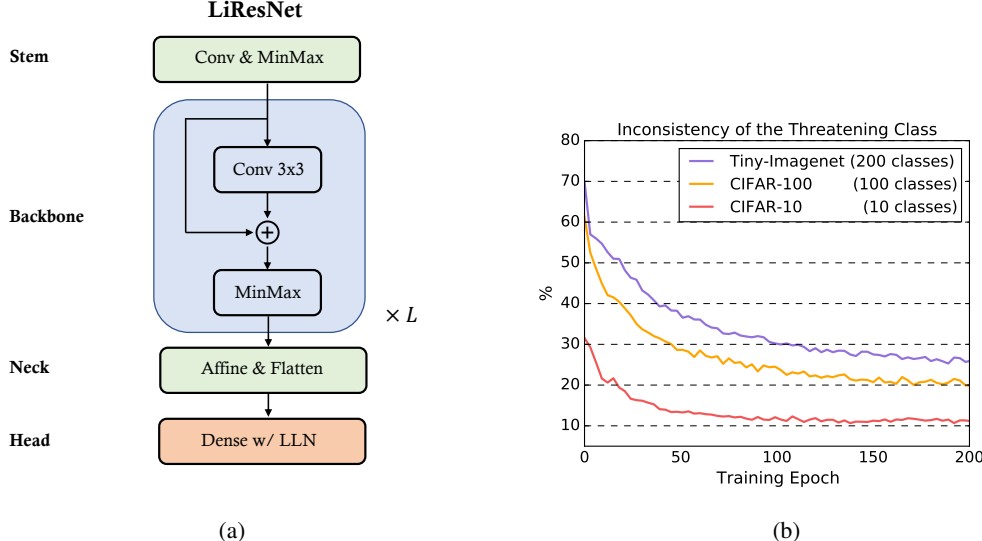

(a)

(b)

Figure 2: **(a)** An illustration of the LiResNet Architecture. **(b)** Plot of the percentage of instances at each epoch during training for which the *threatening* class (the one with the second-highest logit value) differs compared to the previous epoch (on the same instance). Numbers are reported on three datasets with 10, 100, and 200 classes using a GloRo LiResNet with GloRo TRADES [32] loss.

**Robustness Needs More Capacity.** Despite the fact that robust and correct predictions, i.e., VRA=100%, is achievable for many standard datasets [55], the state-of-the-art VRAs (which are achieved using Lipschitz-based certification) are far away from realizing this goal. Recent works have emphasized the role of network *capacity* in learning robust classifiers, suggesting a lack of capacity may be a factor in this shortcoming, beyond the more obvious fact that Lipschitz-based certification is conservative and may falsely flag robust points. Bubeck and Sellke [4] showed that a smooth (and thus robust) decision boundary requires $d$ times more parameters than learning a non-smooth one, where $d$ is the ambient data dimension. In addition, to *tightly certify* a robust boundary with Lipschitz-based approaches, Leino [30] demonstrated the need for extra capacity to learn smooth level curves around the decision boundary, which are shown to be necessary for tight certification.

**Robust Residual Blocks.** Prior work [22] has introduced residual connections, which have proven effective for scaling network depth. Residual connections have since become one of the building blocks for Transformer networks [51]. High-capacity ResNets and Vision Transformers (ViT) [14] outperform basic ConvNets in terms of *clean accuracy* (i.e., ignoring robustness) on major research datasets, but similar success has not been observed for VRAs. For example, several papers find ResNets have lower VRAs on CIFAR-10 compared to ConvNets [42, 48] while no paper has reported VRAs on common vision datasets with ViTs. These empirical findings are seemingly at odds with the capacity arguments raised in the literature, however this may be due to unfavorable training dynamics of current approaches. This work aims to investigate the reasons behind this discrepancy to capture the potential of residual connections. Our primary focus is the residual connection (to skip convolutions) in ResNets; however, the implications of this work include paving the way for future work on certifying other types of residual connections, e.g. skipping attention in Transformers.

## 3 Linear-Residual Networks

The GloRo approach can easily be adapted from a ConvNet architecture to a ResNet architecture by simply adjusting how the Lipschitz constant is computed where the residual and skip paths meet. Consider a conventional residual block $r(x)$ given by $r(x) = x + g(x)$, where the residual branch, $g$, is a small feed forward network, typically with 2-3 convolutional/linear layers paired with nonlinear activations. The Lipschitz constant, $K_r$, of $r(x)$, with respect to the Lipschitz constant, $K_g$, of $g(x)$ is upper-bounded by $K_r \leq 1 + K_g$.

Thus, a typical estimation of the residual block's Lipschitz constant is $1 + K_g$. However, this is a loose estimation. To see this, let $u = \arg\max_x \|r(x)\|/\|x\|$, and observe that the bound is only tight if $u$ and $r(u)$ point in the *same direction* in the representation space. This is presumably unlikely to happen, as random vectors are almost orthogonal in high-dimensional space with high probability. Thus, using $1 + K_g$ as the Lipschitz constant of $r(x)$ is unlikely to be tight even if $K_g$ is tight.

**The LiResNet Architecture.** As discussed, the standard residual block is fundamentally challenging to obtain a tight Lipschitz bound on. Thus, we propose to use a *Linear Residual Block*:

$$r_{\mathsf{linear}}(x) = x + \mathrm{Conv}(x)$$

Since $r_{\mathsf{linear}}(x)$ is still a linear transformation of $x$, we can easily compute the equivalent convolution which has the same output as $r_{\mathsf{linear}}(x)$. For example, consider a convolution layer with $W \in \mathbb{R}^{(2k_0+1) \times (2k_0+1) \times n \times n}$ where $(2k_0 + 1)$ is the kernel size and $n$ is the number of channels and zero padding $k_0$. The weights of the equivalent convolution are then $W + \Delta$ where $\Delta[k_0, k_0, i, i] = 1$ for all $i \in \{1, \cdots, n\}$ and all other entries are set to zero. Thus, the Lipschitz constant of $r_{\mathsf{linear}}$ can be efficiently estimated using the power method [16]. Nonlinearity of the network is then obtained by adding nonlinear activations (e.g., MinMax [1]) to the outputs of each residual block. By stacking multiple linear residual blocks (with interjecting activations), and including a stem, neck, and head, we obtain the *Linear ResNet* (LiResNet) architecture, illustrated in Figure 2a.

**A Note on the Power of LiResNet.** It is worth noting that, despite the apparent simplicity of linear residual blocks, the LiResNet architecture is surprisingly powerful. First, although $r_{\mathsf{linear}}$ can be parameterized as a single convolution, linear residual blocks are meaningfully different from convolutional layers. Specifically, linear residual blocks contain residual connections which facilitate better gradient propagation through the network. As an analogy, convolutional layers can be parameterized as dense layers that share certain weights (and zero others out), but convolutions provide meaningful regulation not captured in a standard dense layer. Similarly, linear residual blocks provide meaningful regularization that improves training dynamics, especially in deep models.

Second, while the linearity of linear residual blocks may seem to limit the expressiveness of the LiResNet architecture, this intuition is perhaps misleading. A useful analogy is to think of a ResNet with a fixed number of total layers, where we vary the number of layers per block. On one extreme end, all the layers are in a single block, resulting in a simple CNN architecture. A typical ResNet has a block size of 2 or 3; and, while adding more blocks imposes regularization that could reduce the model's expressiveness, this is not generally considered a disadvantage for ResNets, since the residual connections admit much deeper models, which ultimately makes up for any capacity that would be lost. LiResNets can be seen as the other extreme end, where each block has a single layer. While this entails further regularizaiton the same principle holds pertaining to the depth of the model.

**A Note on Transformer-based Architectures.** Because Transformer-based models also consist of many residual blocks, e.g., skip connections in the multi-head self-attention modules, we have experimented if the proposed linear residual block helps to certify Transformer-based models. Our primary finding is, while the linear residual block tightens the Lipschitz constant of the skip connection, the attention module is another (bigger) bottleneck for certifying Transformers. Namely, self-attention is not a Lipschitz function [28], which is incompatible with Lipschitz-based certification approaches. Recent works [9, 28] provide a few Lipschitz-continuous alternatives for attention modules, none of which are found to have competitive certifiable robustness compared to LiResNets in our experiments. Thus, the evaluation part of this paper (Section 5) will mainly focus on comparing LiResNets with prior work, while our experiments with Transformer-based models will be included in Appendix I for reference in future work.

## 4    Efficient Margin Maximization

Using the LiResNet architecture, we are able to train far deeper models. However, we observe that the GloRo cross-entropy loss (and the GloRo TRADES loss variant) used by Leino et al. [32], which we will refer to as *standard* GloRo loss, becomes inefficient as the number of classes in the dataset increases. Standard GloRo loss minimizes the logit score of the $\perp$ class (i.e. $f_\perp$ in Eq. 1), which only affects one margin—namely, the one between the predicted class and the *threatening* class (the one with the second-highest logit value)—at each iteration. To see this, we show in Figure 3

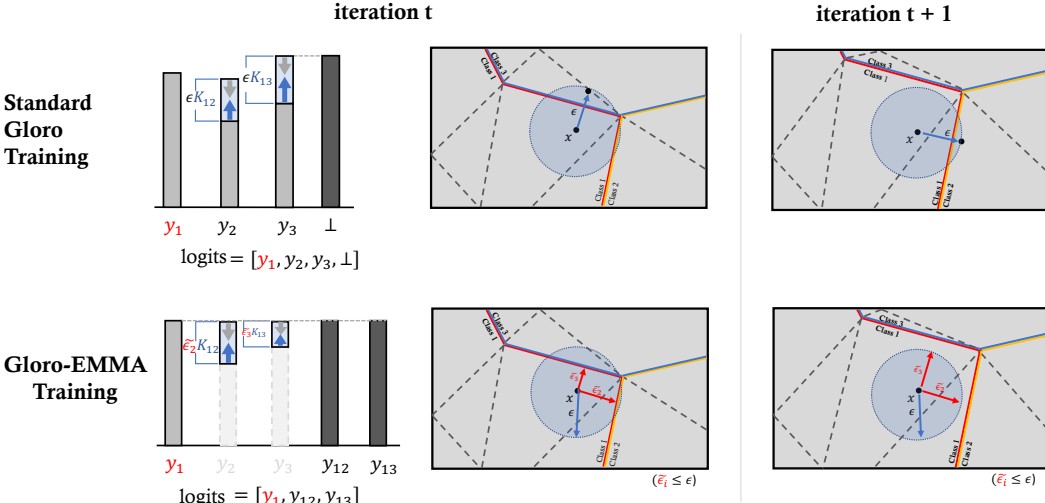

Figure 3: Comparison between the standard GloRo loss and our proposed GloRo EMMA loss (Definition 3). The standard GloRo loss constructs a $\perp$ class to push the nearest decision boundary away while EMMA adds the perturbations to all rest classes (i.e. considering class 1 is the ground truth) to efficiently push all boundaries away. Another difference is that EMMA uses the adaptive $\tilde{\epsilon}_i$ instead of $\epsilon$ in the training. At iteration $t+1$, the network using EMMA loss is already robust while standard GloRo loss may take more iterations to become robust.

that the standard GloRo loss largely focuses on the decision boundary between classes 1 and 3 at iteration $t$, even though class 2 is also competitive but slightly less so than class 3. Furthermore, the possibility arises that the threatening class will alternate between competing classes leading to a sort of "whack-a-mole" game during training.

Indeed, we find this phenomenon to be increasingly common as the number of classes grows. Figure 2b shows the fraction of instances at each epoch for which the threatening class differs from in the preceding epoch. In the initial 100 epochs, more than 30% instances from Tiny-ImageNet (containing 200 classes) have different threatening classes at each each iteration, while the same number for CIFAR-10 is only about 10%. At best, this contributes to making training with many classes less efficient, and at worst, it halts progress as work in one epoch is undone by the next.

To address this problem, we propose a new loss function known as the *Efficient Margin Maximization* (EMMA) loss (Definition 3) and we consequently refer to this method as *GloRo (EMMA)*. Conceptually, the EMMA loss function adds the maximum possible margin to each non-ground-truth class. This margin is defined as the maximum gain a given logit can achieve over the ground truth logit within the *current* largest $\ell_p$-ball.

Formally, suppose $f_i$ denotes the logit of class $i$ and $K_{yi}$ is the margin Lipschitz between class $i$ and label $y$ as defined in Equation 1. We can express the radius of the *current* largest certifiable $\ell_p$-ball for a class $i$ relative to the ground-truth class $y$ as $\kappa_i$. This value is determined by their logit margin over the Lipchitz constant of this margin, as shown in Equation 2.

$$\forall i, \kappa_i(x) = \frac{f_y(x) - f_i(x)}{K_{yi}} \text{ if } i \neq y \text{ and } 0 \text{ otherwise.} \tag{2}$$

Taking into account that the expected robust radius is $\epsilon$, we formulate EMMA loss as follows.

**Definition 3** (Efficient Margin MAximization (EMMA) loss). Suppose the radius of the current $\ell_p$-ball the model $F$ can certify robustness respects to ground truth class $y$ over class $i$ is $\kappa_i(x)$ as in Equation 2. We define EMMA loss for data $(x, y)$ as follows,

$$\ell_{\text{EMMA}} = -\log \frac{\exp(f_y(x))}{\sum_i \exp[f_i(x) + \tilde{\epsilon}_i K_{yi}]} \text{ and } \tilde{\epsilon}_i = \texttt{clip\_value\_no\_grad}(\kappa_i(x), 0, \epsilon).$$

When using EMMA loss, the actual radius at which we require the model to certify robustness at each iteration is $\tilde{\epsilon}_i$ for the margin between class $i$ and $y$. It's important to note that if the model still

Table 1: Comparing the proposed architecture LiResNet using the proposed loss EMMA against baseline methods. Verifiablly Robust Accuracy (VRA) is the percentage of test points on which the model is both correct and certifiablly robust so a higher VRA is better. An $\ell_2$-ball with a radius of $\epsilon = {}^{36}\!/_{255}$ is used for all experiments following the conventions in the literature. "+DDPM" means we use DDPM generators [36] to augment the training set (no external data used).

| Dataset | Method | Architecture | Specification | #Param. (M) | Clean (%) | VRA (%) |
|---|---|---|---|---|---|---|
| **CIFAR-10** | BCOP [33] | ConvNet | 6C2F | 2 | 75.1 | 58.3 |
| | GloRo [32] | ConvNet | 6C2F | 2 | 77.0 | 60.0 |
| | Local-Lip Net [24] | ConvNet | 6C2F | 2 | 77.4 | 60.7 |
| | Cayley [48] | ConvNet | 4C3F | 3 | 75.3 | 59.2 |
| | SOC (HH+CR) [43] | ConvNet | LipConv-20 | 27 | 76.4 | 63.0 |
| | CPL [35] | ResNet | XL | 236 | 78.5 | 64.4 |
| | SLL [2] | ResNet | XL | 236 | 73.3 | 65.8 |
| | GloRo + EMMA (ours) | LiResNet | 6L128W | 5 | 78.7 | 64.4 |
| | GloRo + EMMA (ours) | LiResNet | 12L512W | 49 | 81.3 | **66.9** |
| | + DDPM | LiResNet | 12L512W | 49 | 82.1 | **70.1** |
| **CIFAR-100** | BCOP [33] | ConvNet | 6C2F | 2 | 45.4 | 31.7 |
| | Cayley [48] | ConvNet | 4C3F | 3 | 45.8 | 31.9 |
| | SOC (HH+CR) [43] | ConvNet | LipConv-20 | 27 | 47.8 | 34.8 |
| | LOT [54] | ConvNet | LipConv-20 | 27 | 49.2 | 35.5 |
| | SLL [2] | ResNet | XL | 236 | 46.5 | 36.5 |
| | GloRo + EMMA (ours) | LiResNet | 6L128W | 5 | 52.1 | 36.3 |
| | GloRo + EMMA (ours) | LiResNet | 12L512W | 49 | 55.2 | **38.3** |
| | + DDPM | LiResNet | 12L512W | 49 | 55.5 | **41.5** |
| **Tiny-ImageNet** | GloRo [32] | ConvNet | 8C2F | 2 | 35.5 | 22.4 |
| | Local-Lip Net [24] | ConvNet | 8C2F | 2 | 36.9 | 23.4 |
| | SLL [2] | ResNet | XL | 236 | 32.1 | 23.2 |
| | GloRo + EMMA (ours) | LiResNet | 6L128W | 5 | 40.8 | 29.0 |
| | GloRo + EMMA (ours) | LiResNet | 12L512W | 49 | 44.6 | **30.6** |
| | + DDPM | LiResNet | 12L512W | 49 | 46.7 | **33.6** |
| **ImageNet** | GloRo + EMMA (ours) | LiResNet | 12L588W | 86 | 45.6 | 35.0 |

predicts class $i$ over the ground truth class $y$ (i.e., $f_y(x) < f_i(x)$ thus $\kappa_i(x) < 0$), we clip $\tilde{\epsilon}_i$ to 0 and EMMA loss reduces to a non-robust cross-entropy loss (for class $i$). In this case, the training focuses on improving clean accuracy first. As the training progresses, $\tilde{\epsilon}_i$ may grow to the same magnitude as the expected radius $\epsilon$. Once this happens, we shift the model's focus towards certifying other instances, rather than continuously increasing the robust radius for $x$.

As an implementation detail, we need to stop the gradient back-propagation from $\tilde{\epsilon}_i$, i.e., we treat it as a constant during optimization. Otherwise, the denominator term $f_i(x) + \tilde{\epsilon}_i K_{yi}$ is just $f_y(x)$ and the logit for class $i$ is removed from the computation graph of the loss.

Notice that EMMA introduces a dynamic adjustment to the robust radius using $\tilde{\epsilon}_i$. This allows the strength of the robustness regularization to adapt in accordance with the training progress. Specifically, EMMA initially encourages the model to make *correct* predictions when $\tilde{\epsilon}_i = 0$, then it gradually shifts the model towards making more *robust* predictions as $\tilde{\epsilon}_i \to \epsilon$. The idea of balancing the standard accuracy and robustness perhaps also agrees with the design of TRADES (for adversarial training) [59] and its GloRo counterpart [32]. Further discussion on this point is included in Appendix A.

Notice also that the proposed EMMA loss only affects the training step, so we still use Equation 1 to construct the $\perp$ logit to certify the local robustness during inference, as proposed by Leino et al. [32].

## 5 Evaluation

In this section, we provide an empirical evaluation of LiResNet and EMMA loss in comparison to certifiably robust training approaches in prior works. We begin by comparing the best VRAs we achieve against the best VRAs reported in the literature in Section 5.1. Next, in Section 5.2, we run head-to-head comparisons between EMMA and the standard GloRo losses as an ablation study to measure the empirical benefits of EMMA loss. Section 5.3 presents experiments on networks of different depths, to shed light on the unique depth scalability of the LiResNet architecture. In

Appendix F, we dive deeper into the ImageNet results, providing another novel observation regarding the impact of the number of classes on VRA.

The models in our evaluation either follow the architecture choices of prior work, or belong to a family of LiResNet architectures shown in Figure 2a. Because the stem, neck, and head are essentially fixed in our LiResNet architectures, we refer to a particular architecture variation by the number of backbone blocks, **L**, and the number of input/output channels, **W**, in convolution layers of each backbone. For example, **6L128W** refers to a LiResNet with 6 linear residual blocks with 128 channels each. Other architecture specifications can be found in Appendix B.

## 5.1 Improved VRA with LiResNet

We compare GloRo LiResNets trained with EMMA loss with the following baselines from the literature: GloRo Nets with ConvNet architecture and cross-entropy/TRADES loss [32], BCOP [33], Cayley [48], Local-Lip Net [24], and SOC with Householder and Certification Regularization (HH+CR) [43], CPL [35] and SLL [2], which are selected for having been shown to surpass other approaches. We experiment on CIFAR-10/100 and Tiny-ImageNet using $\ell_2$ perturbations with $\epsilon = {}^{36}/_{255}$, the standard datasets and radii used in prior work. Additionally, we demonstrate the scalability of GloRo LiResNets by training on and certifying ImageNet. We report the clean accuracy, VRA, and the model size (by # of parameters) in Table 1. Further details on training are in Appendix B.

On CIFAR-10/100 and Tiny-ImageNet, we find that a small LiResNet, 6L128W, is able to match or outperform all baselines. This architecture is about *40×* smaller than the largest[4] and highest-performing baseline model (SLL), but comes close to the same accuracy on CIFAR-10/100, and outperforms on Tiny-Imagenet. Our larger LiResNet (12L512W) surpasses all prior approaches, setting the state-of-the-are VRAs (without DDPM augmentation) to 66.9%, 38.3% and 30.6% on CIFAR-10, CIFAR-100 and Tiny-ImageNet, respectively.

Recent work on promoting empirical robustness has found that augmenting the training set with examples generated by *Denoising Diffusion Probabilistic Models* (DDPMs) [23, 36] can further boost the *empirical* robust accuracy [20]. Thus, we also experiment to see if the same success can be found for *certifiable* robustness. DDPMs are trained with the training set, so the generated images do not leak information of the test set and do not use external data. Additional rows with "+DDPM" in Table 1 show that with this augmentation method we further improve VRAs to 70.1%, 41.5% and 33.6% on CIFAR-10, CIFAR-100 and Tiny ImageNet, respectively (further details in Appendix H). Because diminishing returns have been observed for improving empirical robustness when augmenting the training set of ImageNet with DDPM data [20], we did not use DDPM for LiResNet on ImageNet.

**Scaling to ImageNet with GloRo.** There is no theoretical limitation to the other approaches considered in our evaluation that limit us from training them on ImageNet; however, practical resource constraints prevent us from training until convergence with non-GloRo approaches. For example, baselines using orthogonalized kernels—e.g., Cayley, BCOP, and SOC—do not easily fit into memory with $224 \times 224 \times 3$ images, and local Lipschitz computation—e.g., Local-Lip Net—is both time and memory intensive. To the best of our knowledge, we are the first to report the VRA on ImageNet with a *deterministic* robustness guarantee and the first to calculate that VRA using the entire validation set, in contrast to the prior work which only evaluates 1% [25, 39, 40] or 2% [5] of the validation set using Randomized Smoothing [6] (also because of efficiency concerns).

## 5.2 Improved VRA with EMMA

This section demonstrates the empirical gains obtained by switching the typical loss used with standard GloRo losses— i.e., GloRo cross-entropy (CE) and GloRo TRADES—to EMMA. For both ConvNets and LiResNets, we experiment on CIFAR-10, CIFAR-100 and Tiny-ImageNet and report

---

[4]We observe that the SLL XL models have orders-of-magnitude more parameters than the other baselines, coming from a non-standard architecture with many wide dense layers. When we use the 12L512W architecture but with the LiResNet layers replaced with SLL layers, we find the performance is much lower (59.5% VRA on CIFAR-10 and 30.2% VRA on CIFAR-100), suggesting that the primary factor in performance for SLL XL is the large number of parameters rather than the approach to bounding the Lipschitz constant of the residual layers. Capacity has been shown to be an important factor for obtaining high VRA with Lipschitz-based approaches [30].

| Architecture | dataset | TRADES | EMMA | | Dataset | $L$ | ConvNet | ResNet | LiResNet |
|---|---|---|---|---|---|---|---|---|---|
| | CIFAR-10 | 58.8 | 59.2 | | | 6 | 64.0 | 60.3 | 65.5 |
| ConvNet | CIFAR-100 | 34.0 | 35.0 | | CIFAR-10 | 12 | 59.2 | 60.0 | 66.3 |
| | Tiny-ImageNet | 26.6 | 27.4 | | | 18 | $\times$ | 60.1 | 66.6 |
| | CIFAR-10 | 66.2 | 66.3 | | | 6 | 36.5 | 33.5 | 37.2 |
| LiResNet | CIFAR-100 | 37.3 | 37.8 | | CIFAR-100 | 12 | 35.0 | 33.5 | 37.8 |
| | Tiny-ImageNet | 28.8 | 30.3 | | | 18 | $\times$ | 33.6 | 38.0 |
| (a) | | | | | (b) | | | | |

Table 2: **(a)** VRA performance (%) of a ConvNet and a LiResNet on three datasets with different loss functions. **(b)** VRA (%) performance on CIFAR-10/100 with different architectures ($L$ is the number of blocks in the model backbone). We use EMMA loss for Gloro training. All models in this table use 256 channels in the backbone. A value of $\times$ indicates that training was unable to converge.

VRAs in Table 2a. The ConvNets we use are modified from the baseline 6C2F architecture used by Leino et al. [32] to have the same (wider) channel width as our 6L256W architecture, which is why we achieve higher VRAs than originally reported in [32]. The remaining implementation details and the clean accuracy of each model can be found in Appendix C.

We see in Table 2a that the performance gain from switching TRADES to EMMA becomes clear when the number of classes increases from 10 (CIFAR-10) to 200 (Tiny-Imagenet). This observation aligns with our hypothesis used to motivate EMMA loss, discussed in Section 4, namely, that the rotating threatening class phenomenon observed during training (see Figure 2b) may contribute to suboptimal learning.

## 5.3 Going Deeper with LiResNet

As depicted in Figure 1 from the introduction, the VRA obtained using GloRo LiResNets scales well as the depth of the model increases, while prior work has failed to further improve the best achievable VRA through additional layers. To further validate that the ability to successfully go deeper primarily comes from the structural improvement of linear residual branch in the LiResNet architecture—as opposed to being an advantage of the framework, GloRo, itself—we run head-to-head comparisons on CIFAR-10 and CIFAR-100 of GloRo Nets using (1) a feed-forward ConvNet architecture, (2) a conventional ResNet architecture, and (3) a LiResNet architecture. We train all three architectures with EMMA loss at three different depths. We report VRAs of these models in Table 2b (implementation details are given in Appendix D).

We see in Table 2b that very deep ConvNets may not be able to converge even on small-scale datasets like CIFAR-10 and CIFAR-100. Moreover, the VRA of both ConvNets and conventional ResNets do not benefit from the increasing network depth—in fact performance *decreases* as the network is made significantly deeper. By contrast, LiResNet is the only architecture under the same conditions that benefits from more layers, showing its unique promise for scalability. In Appendix D, we include more results with even deeper LiResNets on CIFAR-10, CIFAR-100 and Tiny-ImageNet.

## 6 Related Work and Discussion

**Tightening Lipschitz Bound.** Work closely related to enhancing VRA with architectural redesign includes the use of orthogonalized convolutions [2, 35, 42, 48, 54], which are 1-Lipschitz by construction. In a similar vein, we introduce the linear residual branch to solve the overestimation of Lipschitz Constant in the conventional ResNet. Our linear residual layer compares favorably to orthogonalized layers with a few key advantages. Specifically, although there is no technical limitation that would prevent us from using orthogonalized convolutions in LiResNets, GloRo regularization performs better in our experiments, and is significantly less expensive than training with orthogonalized kernels.

**Model Re-parameterization.** The LiResNet block is re-parameterization of a convolution layer, to make it trainable at large depth. Prior to this work, there are seveval studies use the same technique: DiracNet [58], ACB [11], RepVGG [12], RepMLP [13], etc. Our work has different motivations

from these works, which use re-parameterization to achieve better optimization loss landscapes. Our work follows a similar approach to obtain tighter estimates of Lipschitz bounds.

**Towards Certifying Transformers.**   Although our current experiments (see Appendix I) indicate that Vision Transformers—utilizing linear residual blocks and Lipschitz-attention [9, 28]—do not yield VRAs comparable to those of LiResNets, we conjecture this discrepancy primarily arises from the loose Lipschitz upper-bounds inherent to most Lipschitz-attention mechanisms, in contrast to the *exact* Lipschitz constants of convolutions in LiResNets. To fully harness the potential of the linear residual block, future work should focus on *tightening* the Lipschitz bound for Lipschitz-attention.

**Randomized Smoothing.**   As opposed to the deterministic robustness guarantee focused on in this work, probabilistic guarantees, based on *Raondmized Smoothing* (RS) [6], have been long studied at ImageNet-scale [5, 25, 39, 40]. Despite RS's reported higher certification results on ImageNet compared to GloRo LiResNet for the same $\epsilon$, it has two primary limitations. First, RS can provide false positive results, where adversarial examples are given robustness certificates. This, in many real-world security and financial contexts, can be untenable, even at a 0.1% false positive rate (FPR). Additionally, the computational cost of RS-based certification is orders of magnitude higher than Lipschitz-based certification—to certify one instance with FPR=0.1%, RS requires 100,000 extra inferences, and this number increases exponentially for lower FPR [6]. This resource-consuming nature of RS-based certification limits the type of applications one can deploy it in. Even in academic research, methods on certifying ImageNet with RS only report results using 1% of the validation images (i.e. 500 images) [25, 39, 40]; however, with the Lipschitz-based approach employed in our work, we can certify the entire validation set (50,000 images) in *less than one minute*.

**Limitations for Robust Classification.**   The development of EMMA and empirical studies on class numbers' impact on VRA highlight the challenge of scaling robustness certification to real-world datasets. One challenge is that an increasing number of classes places increasing difficulty on improving VRA; we provide more details on this in Appendix F. In addition, ambiguous or conflicting labels can further exacerbate the issue, indicating a possible mismatch between the data distribution and the objective of *categorical accuracy*, especially in its robust form. A number of mislabeled images have been identified in ImageNet [3, 37, 50, 57], raising obstacles for robust classification. In addition, real-world images often contain multiple semantically meaningful objects, making robust single-label assignment problematic and limiting the network's learning potential unless alternate objectives (e.g., robust top-$k$ accuracy [31] and robust segmentation [17]) are pursued.

## 7   Conclusion

In this work, we propose a new residual architecture, LiResNet, for training certifiably robust neural networks. The residual blocks of LiResNet admit tight Lipshitz bounds, allowing the model to scale much deeper without over-regularization. To stabilize robust training on deep networks, we introduce Efficient Margin Maximization (EMMA), a loss function that simultaneously penalizes worst-case adversarial examples from all classes. Combining the two improvements with GloRo training, we achieve new state-of-the-art robust accuracy on CIFAR-10/100 and Tiny-ImageNet under $\ell_2$-norm-bounded perturbations. Furthermore, our work is the first to scale up deterministic robustness guarantees to ImageNet, showing the potential for large scale deterministic certification.

## Broader Impact

Machine learning's vulnerability to adversarial manipulation can have profound societal implications, especially in applications where robust and reliable AI systems are paramount. This work is an important step towards increasing the robustness of neural networks to adversarial attacks, and more broadly, towards ensuring the trustworthiness of AI systems. Our ability to scale up fast deterministic robustness guarantees to ImageNet—a dataset more reflective of the complexity and diversity of real-world images—indicates that our approach to robust learning can be applied to practical, real-world applications. Nevertheless, while these advancements are promising, they also emphasize the need for ongoing vigilance and research in the face of increasingly sophisticated adversarial attacks. Ensuring that AI systems are robust and trustworthy will remain a critical task as these technologies continue to permeate society.

# Acknowledgements

The work described in this paper has been supported by the Software Engineering Institute under its FFRDC Contract No. FA8702-15-D-0002 with the U.S. Department of Defense, as well as DARPA and the Air Force Research Laboratory under agreement number FA8750-15-2-0277.

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

# A  Discussions on EMMA Loss

One important aspect of EMMA loss is the dynamic adjustment to the robustness radius for each class when determining the loss. The robust radius used in EMMA is $\tilde{\epsilon}$ instead of the radius $\epsilon$ that is used at run time. By the definition of EMMA, $0 \le \tilde{\epsilon} \le \epsilon$ and $\tilde{\epsilon}$ is 0 if the model is not labeling the input correctly. $\tilde{\epsilon}$ grows as the model becomes more robust at the corresponding input. As a result, when the model is not sufficiently robust at the input, EMMA uses $\tilde{\epsilon} < \epsilon$ and imposes milder Lipschitz regularization. On the other hand, if using a fixed margin, the loss function turns out to be:

$$\ell_{\text{fixed}} = -\log \frac{f_y(x)}{\sum_i f_i(x) + \epsilon K_{yi}} \tag{3}$$

This bears similarities to Lipschitz Margin loss, which has been used in prior work for certified training [49], although Lipschitz Margin loss typically uses $\sqrt{2}\epsilon K$, which gives a looser approximation than directly using the Lipschitz constant for each margin, i.e., $K_{yi}$. The fixed margin loss $\ell_{\text{fixed}}$ in Equation 3 penalizes the Lipschitz-adjusted margin between the ground truth class and all other classes. Therefore, this loss function imposes a stronger regularization on the Lipschitz constant of the model than EMMA loss, and limits the model capacity more. We find that models trained with the fixed margin loss require weaker data augmentation or smaller training $\epsilon$ to avoid underfitting. However, this can instead lead to robust overfitting. The gap between clean accuracy and VRA is notably higher for models trained with the fixed margin loss, and the overall performance is worse compared to models trained with the dynamic margin loss, i.e., EMMA loss.

# B  Implementation Details for Table 1

## B.1  Training details

**Dataset details**  The input resolution is 32 for CIFAR10/100, 64 for Tiny-ImageNet and 224 for ImageNet respectively. We apply the following data augmentation to CIFAR datasets: random cropping, RandAugment [8], random horizontal flipping. For Tiny-ImageNet, we find this dataset is easy to overfit and add an extra Cutout [10] augmentation. For data augmentation hyper-parameters, we use the default PyTorch setting.

**Platform details**  Our experiments were conducted on an 8-GPU (Nvidia A100) machine with 64 CPUs (Intel Xeon Gold 6248R). Each experiment on CIFAR10/100 and Tiny-ImageNet takes one GPU and each experiment on ImageNet takes 8 GPUs. Our implementation is based on PyTorch [38].

**Training details**  On the first 3 datasets, all models are trained with the NAdam [15] with the Lookahead optimizer wrapper [61] with a batch size of 256 and a learning rate of $10^{-3}$ for 800 epochs. We use a cosine learning rate decay [34] with linear warmup [21] in the first 20 epochs. On ImageNet, we only change the batch size to 1024 and training epochs to 400.

During training, we schedule the training $\epsilon$ to ramp up from small values and slightly overshoot the test epsilon. Let the total number of epochs be $T$ and the test certification radius be $\epsilon$, we use

$$\epsilon_{\text{train}}(t) = \left( \min(\frac{2t}{T}, 1) \times 1.9 + 0.1 \right) \epsilon, \quad \epsilon = 36/255.$$

at epoch $t$. As a result, $\epsilon_{\text{train}}(t)$ begins at $0.1\epsilon$ and increases linearly to $2\epsilon$ before arriving halfway through the training. Later, $\epsilon_{\text{train}}$ remains $2\epsilon$ to the end.

## B.2  Model architecture details

**Model stem** is used to convert the input images into feature maps. On CIFAR10/100, we use a convolution with kernel size 5, stride 2, and padding 2, followed by a MinMax activation as the stem. On Tiny ImageNet, we use a convolution with kernel size 7, stride 4, and padding 3, followed by a MinMax activation as the stem. On ImageNet, we follow the ViT-like patching [14] and use a convolution with kernel size 14, stride 14, and padding 0, followed by a MinMax activation as the stem. Thus the output feature map size from the stem layer is $16 \times 16$ for all 4 datasets. The number of filters used in the convolution is equal to the model width $W$.

Table 3: Clean accuracy and VRA performance (%) of a ConvNet and a LiResNet on three datasets with different loss functions

| loss | TRADES | | EMMA | |
| --- | --- | --- | --- | --- |
| | Clean (%) | VRA (%) | Clean (%) | VRA (%) |
| **CIFAR-10** ($\epsilon = {}^{36}/{}_{255}$, 10 classes) | | | | |
| ConvNet | 71.7 | 58.8 | 72.5 | 59.2 |
| LiResNet | 79.6 | 66.2 | 80.4 | 66.3 |
| **CIFAR-100** ($\epsilon = {}^{36}/{}_{255}$, 100 classes) | | | | |
| ConvNet | 53.4 | 34.0 | 50.6 | 35.0 |
| LiResNet | 57.8 | 37.3 | 54.2 | 37.8 |
| **Tiny-ImageNet** ($\epsilon = {}^{36}/{}_{255}$, 200 classes) | | | | |
| ConvNet | 42.2 | 26.6 | 40.0 | 27.4 |
| LiResNet | 45.8 | 28.8 | 43.6 | 30.0 |

**Model backbone** is used to transform the feature maps. It is a stack of $L$ LiResNet blocks followed by the MinMax activation, i.e., (LiResNet block $\rightarrow$ MinMax) $\times L$. We keep the feature map resolutions and the number of channels constant in the model backbone. We find some tricks in normalization-free residual network studies [41, 60] can improve the performance of our LiResNet as our method is also a normalization-free residual network. Specifically, we add an affine layer $\beta$ that applies channel-wise learnable multipliers to each channel of the feature map (similar to the affine layer of batch normalization) and a scaler of $1/\sqrt{L}$ to the residual branch where $L$ is the number of blocks:

$$y = x + \frac{1}{\sqrt{L}} \beta \text{Conv}(x)$$

**Model neck** is used to convert the feature maps into a feature vector. In our implementation, the model neck is a 2 layer network. The first layer is a convolution layer with kernel size 4, stride 4, and padding 0, followed by a MinMax activation. The number of input channels is the model width $W$ and the number of output channels is $2W$. Then we reshape the feature map tensor into a vector. The second layer is a dense layer with output dimension $d$ where $d = 2048$ for the three small datasets (CIFAR10/100 and Tiny-ImageNet) and $d = 4096$ for ImageNet.

**Model head** is used to make classification predictions. We apply the last layer normalization (LLN) proposed by [43] to the head.

### B.3   Metric details

We report the clean accuracy, i.e., the accuracy without verification on non-adversarial inputs and the verified-robust accuracy (VRA), i.e., the fraction of points that are both correctly classified and certified as robust. Our results are averaged over 5 runs for CIFAR10/100 and TinyImageNet and 3 runs for ImageNet.

## C   Details for Table 2a

In Table 2a, we use an L12W256 configuration, i.e., the backbone has 12 blocks and the number of filters is 256. For ConvNet, the only difference is that the LiResNet block is replaced by a convolution of kernel 3, stride 1, and padding 1. All other settings are the same. Table 3 is a more detailed version of Table 2a with the clean accuracy.

## D   Details for Table 2b

In Table 2b, we use the configuration of W256, i.e., the number of channels in the backbone is 256. The only difference between conventional ResNet and LiResNet is the block. The block for

Table 4: Clean accuracy and VRA (%) performance on CIFAR-10/100 with different architectures ($L$ is the number of blocks in the model backbone). We use EMMA loss for Gloro training. $\times$ stands for not converging at the end.

| Dataset | $L$ | ConvNet | | ResNet | | LiResNet | |
|---|---|---|---|---|---|---|---|
| | | Clean(%) | VRA(%) | Clean(%) | VRA(%) | Clean(%) | VRA(%) |
| | 6 | 77.9 | 64.0 | 74.2 | 60.3 | 79.9 | 65.5 |
| CIFAR-10 | 12 | 72.5 | 59.2 | 74.0 | 60.0 | 80.4 | 66.3 |
| | 18 | $\times$ | $\times$ | 73.9 | 60.1 | 81.0 | 66.6 |
| | 6 | 51.8 | 36.5 | 48.4 | 33.5 | 53.6 | 37.2 |
| CIFAR-100 | 12 | 50.6 | 35.0 | 48.1 | 33.5 | 54.2 | 37.8 |
| | 18 | $\times$ | $\times$ | 48.2 | 33.6 | 54.3 | 38.0 |

Table 5: Clean accuracy and VRA (%) performance of LiResNet of different depths ($L$ is the number of blocks in the model backbone).

| $L$ | CIFAR10 | | CIFAR100 | | Tiny-ImageNet | |
|---|---|---|---|---|---|---|
| | Clean(%) | VRA(%) | Clean(%) | VRA(%) | Clean(%) | VRA(%) |
| 6 | 79.9 | 65.5 | 53.6 | 37.2 | 43.1 | 29.8 |
| 12 | 80.4 | 66.3 | 54.2 | 37.8 | 43.6 | 30.3 |
| 18 | 81.0 | 66.6 | 54.3 | 38.0 | 43.9 | 30.6 |
| 24 | 81.2 | 66.8 | 55.0 | 38.2 | 44.2 | 30.7 |
| 30 | 81.3 | 66.9 | 54.9 | 38.4 | 44.2 | 30.6 |
| 36 | 81.2 | 66.9 | 55.0 | 38.3 | 44.3 | 30.4 |

conventional ResNet is

$$y = x + \beta\mathrm{Conv}(\mathrm{MinMax}(\mathrm{Conv}(x)))$$

where $\beta$ is the affine layer. We find use zeros to initialize $\beta$ works the best for conventional ResNet. The number of input and output channels of the two convolution layers are the same as that of the LiResNet block. Table 4 is a more detailed version of Table 2b with clean accuracy.

## E  Details for Figure 1

We make LiResNet further deeper and study how network depth influences the performance on CIFAR-10/100 and Tiny-ImageNet. Table 5 shows the clean accuracy and VRA of LiResNet (with EMMA loss) on three datasets. All models use a W256 configuration, i.e., the number of convolutional channels is 256. On CIFAR-10/100, the VRA performance of the LiResNet generally improves with depth. On Tiny-ImageNet, the performance remains with the increase of depth.

Figure 1 compares the VRA performance of LiResNet with some existing method for verification robustness on CIFAR-100 (i.e., the 5th of Table 5). The numbers of these methods are taken from their best-reported configurations. The VRA performance of these methods degrades at certain depths, limiting the maximum model capacity of the methods.

## F  Number of Classes vs. VRA

Despite the fact that EMMA loss improves the ability of GloRo Nets to handle learning problems with many classes, datasets with a large number of classes still stand out as particularly difficult for certified training. In principle, a data distribution with less classes is not guaranteed to have more separable features than more classes—indeed, the state-of-the-art clean accuracy for both CIFAR-10 and CIFAR-100 are comfortably in the high 90's despite the large difference in the number of classes. However, training a certifiably robust model with many classes appears more difficult in practice (as observed, e.g., by the large performance gap between CIFAR-10 and CIFAR-100). To test this

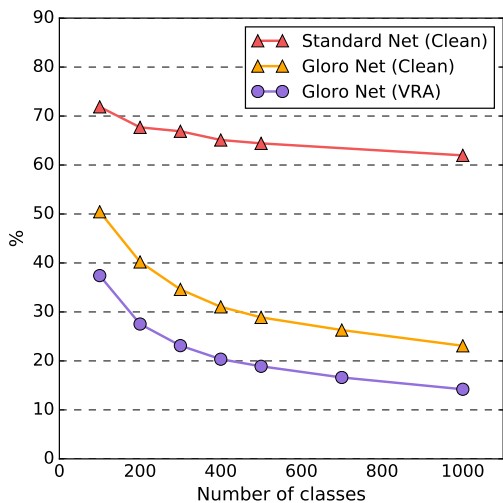

Figure 4: Plot of LiResNet performance on subsets of ImageNet with different number of classes with $\epsilon = 1$

Table 6: VRA (%) of LiResNet of different widths (W).

| W | CIFAR-10 | CIFAR-100 | Tiny-ImageNet |
|---|---|---|---|
| 64 | 64.6 | 36.5 | 28.7 |
| 128 | 65.6 | 37.5 | 29.8 |
| 256 | 66.3 | 37.8 | 30.0 |
| 512 | 66.9 | 38.3 | 30.6 |

observation further, we provide an empirical study on various class-subsets of ImageNet to study the relationship between the number of classes and VRA.

We randomly shuffle the 1000 classes of ImageNet and select the first $100 \cdot k$ classes, where $k \in [10]$, to build a series of subsets for training and testing. For each value of $k$, we train a GloRo LiResNet with EMMA loss ($\epsilon = 1$) and report the clean accuracy and VRA (at $\epsilon = 1$) on the test set. For reference, we also train a standard (i.e., not robust) LiResNet with Cross Entropy and report its clean accuracy on the test set. The final results are shown in Figure 4 with additional details in Appendix F. Compared to the clean accuracy of a standard model, increasing the number of classes leads to a steeper drop in both the VRA and the clean accuracy of the robustly trained models. Specifically, while the performance of the standard model differs only by 10% between a 100-class subset and the full ImageNet, the performance of the GloroNet (both clean accuracy and VRA), drops by 30%.

These results add weight to the observation that, even when mitigated by EMMA loss, large numbers of classes present a particular challenge for certifiably robust learning. This may arise from the need to learn a $2\epsilon$-margin between all regions with different labels, which becomes progressively more challenging as boundaries between a growing number of classes become increasingly difficult to push off the data manifold.

## G    Going Wider with LiResNet

We study how network width (i.e., the number of channels in the model backbone) can influence the performance of LiResNet on CIFAR-10, CIFAR-100 and Tiny-ImageNet. Table 6 shows the results. All models use a L12 configuration. Unlike the network depth, increasing the width can stably improve the model performance within a certain range.

# H  Extra data from DDPM

We use codes from the improved DDPM [36] to train generative models on CIFAR10, CIFAR100 and Tiny-ImageNet. The models are only trained on the training set of each dataset and no external data is used. We use the recommended hyper-parameters from [36] and the models are conditional, i.e., generated samples are with labels. We generate 1 million samples for each dataset.

During the training of Gloro Net, we sample 256 samples from the original dataset and 256 samples from the generated data for each batch. Due to the large total number of generated data, we do not need strong data augmentation on the generated data. Compared to the original dataset, we do not use the RandAugment augmentation for the generated data. All other settings are the same for the original dataset and the generated data.

# I  Certifiable Robustness with Transformers

Transformers-based models [51] have been shown to surpass existing convolution networks on major language tasks. To fully utilize the power of Transformer layers, and Vision Transformers [14] chunk images as small patches to convert one image as a sequence of patches, which is used as input to a Transformer-based network. A Transformer layer is a combination of self-attention blocks (SA) and feed-forward layers (FFN). Between the blocks, layer normalization layers are added to increase the stability of the learning. Because in this paper we mainly focus on certifying vision models, we will simplify refer Vision Transformers as Transformers in the rest of this section.

**Non-Lispchitz Operations in Transformers.**  The fundamental challenge of certifying Transformers' prediction on images with Lipschitz-based approaches arises from the many modules in a Transformer layer are not Lipschitz-continuous, i.e. SA and Layer Normalization. One can simplify remove normalization layers from Transformer layer but this may lead to a serious performance degradation [56]. On the other hand, SA is also not Lipschitz because of the softmax calculation taken over the attention scores. Recent work focuses on designing Lipschitz-continuous alternatives for SAs, which includes OLSA [53] and L2-MHA [28]. Another idea is to use spatial MLP [47] that calculates static "attention weights" so they are bounded by construction.

**Replacing with Linear Residual Connections.**  In terms of FFN, it is a standard residual block and thus suffers from the same problem identified in Section 3. Namely, the overall Lipschitz constant of FFN can be loose because of the skip connection. We therefore replace a standard FFN in a Transformer layer with our linear residual connection to tighten its Lipschitz bound.

**Certifying Transformers.**  Our discussion above has shown that to certify the robustness of Transformer-based models, one needs to remove non-Lipschitz-continuous operations or replace them with Lipschitz-continuous alternatives. We will refer to a Transformer-based model modified to only contain Lipschitz-continuous operations as *Lipschitz Transformers*. We conduct experiments on CIFAR10 and CIFAR100 and provide VRAs of Lipschitz Transformers with different linear operations. That is, for the SA block, we replace it either with an OLSA or a spatial MLP. For the FFN layer, we can optionally it to our linear residual connection and will denote this modified FFN as LiFFN. We use the L12W256 configuration and Table 7 shows the results.

**Results.**  As shown in Table 7, all variants of Lipschitz Transformers do not outperform LiResNets (with the same configuration). The combination of "Spatial MLP" and "LiFFN" performs the best among all variants of transformers. This architecture is also most aligned with the motivation of this paper – using linear operations for the weights to obtain tight Lipschitz constant estimation.

In conclusion, Transformers have potential applicability in Lipschitz-based robustness certification; however, their performance leaves much to be desired. This inadequacy stems primarily from two limitations. Firstly, layer normalization layers are nonviable, and secondly, the SA and FFN blocks struggle to attain a tight estimation of the Lipschitz constant. Although our work sheds a light on how to tightening the Lipschitz bound for FFN with LiFFN, the current Lipschitz alternatives for SAs are not tight enough to further promote the robustness of Transformers.

Table 7: VRAs (%) measured with different variants of Lipschitz Transformers. As a reference for the state-of-the-art performance for convolutional networks, we include our LiResNet results from Table 1.

| dataset | *replacement for* SA | *replacement for* FFN | VRA(%) |
|---|---|---|---|
| CIFAR10 | Spatial MLP | LiFFN | 63.3 |
| | Spatial MLP | FFN | 62.6 |
| | OLSA | FFN | 56.6 |
| | LiResNet-L12W256 | | 66.9 |
| CIFAR100 | Spatial MLP | LiFFN | 36.5 |
| | Spatial MLP | FFN | 33.7 |
| | OLSA | FFN | 28.4 |
| | LiResNet-L12W256 | | 38.3 |

