# OpenReview forum: "Unlocking Deterministic Robustness Certification on ImageNet"
_NeurIPS.cc/2023/Conference — NeurIPS 2023 poster_

### Official Review · Reviewer_YimP · 2023-07-03

**Soundness:** 2 fair
**Presentation:** 2 fair
**Contribution:** 2 fair
**Rating:** 6
**Confidence:** 5

**Summary:**

This paper addresses the issue of certified accuracy with deterministic approaches by utilizing the Lipschitz property of neural networks. The authors propose a novel layer called LiResNet, which enables easy computation of the Lipschitz constant. Additionally, they introduce a new loss function called Efficient Margin Maximization (EMMA), that stabilizes robust training by simultaneously penalizing worst-case adversarial examples from all classes. Finally, the authors conduct experiments on CIFAR-10/100, Tiny-ImageNet, and ImageNet datasets, demonstrating that their approach achieves state-of-the-art results in certified accuracy under $\ell_2$ perturbations and $\epsilon = 36/255$.


**Strengths:**

- The paper is clear and well-written.
- Improving certified accuracy with deterministic approaches is clearly an important problem and the improvement over state-of-the-art is impressive.


**Weaknesses:**

While the experimental results are good and interesting. The paper suffers from overclaiming important contributions and not discussing important related works.

1. **Overclaim on Lipschitz Residual layer**: The claim that the paper is the first to propose a Lipschitz Residual layer is incorrect. The CPL layer [1], and more recently the SLL layer [2], are two previous works that have introduced a 1-Lischitz _Residual_ layer with ReLU activation. The authors mention these two papers in the related work without discussing their contribution. The authors should discuss these works.

2. **On the efficiency and tightness of the approach**: In the abstract, the authors have written: "We show that fast ways of bounding the Lipschitz constant for conventional ResNets are loose, and show how to address this by designing a new residual block"
- **On the efficiency of the approach**: I don't see how the approach is more efficient than previous works, since the authors just use the power iteration (verified in the code provided in the supplementary material as this is not mentioned in the paper or the appendix). The power iteration has been used in three previous works [3, 4, 1]. We can also note that the authors use a power iteration with 10 iterations (default value in the code) while [3,4,1] showed that using only 1 iteration was sufficient and more efficient. Therefore, I don't see how the author’s approach is "more efficient" than previous work.
- **On the tightness of the bound**: It is true that the value of the Lipschitz constant of LiResNet is tighter than the one from a Residual Layer with nonlinearity, in fact, the power iteration computes the _exact_ Lipschitz of LiResNet, since LiResNet is a _linear_ layer -- the PM computes the Lipschitz of the map $x \mapsto (I + W) x$. Computing the Lipschitz of a Residual layer with a nonlinearity leads to a looser bound, because the nonlinearity allows for a lower Lipschitz. The authors should clarify this.

3. **On the new EMMA loss**. The Efficient Margin MAximization (EMMA) loss introduced in the paper is very similar to the one provided by [3]. The authors discuss the difference between their loss and the one from [3] in Appendix A, arguing that the main difference is the use of the _Lipschitz constant for each margin_ while [3] uses the _global Lipschitz constant_ (I believe this paragraph deserves to be in the main paper). This is true, however, the use of the Lipschitz constant for each margin has been proposed twice before [5, 6] (the last layer normalization reduces to the same Lipschitz), therefore, the EMMA loss is a simple combination of two known approaches.

4. **On the experiments**: Table 1 presents results only for the perturbation level of $\epsilon = 36/255$. The authors should present results for a multitude of values to show the overall robustness of their approach (or make a graph with certified accuracy vs \epsilon). A large body of work [1, 2, 6, 7, 8, 9, 10] has presented the certified robustness of their model for at least 3 or 4 perturbation thresholds. I would like to see the same comparison. In Table 2, the authors talk about "VRA performance (%)". What does VRA performance (%) mean? Again the authors should provide VRA with respect to a specific perturbation and not assert that the overall robustness of a model can be computed with only one single perturbation threshold.

**Conclusion**: The paper feels like a patchwork of several existing ideas and looks more like engineering work than research work. The paper has combined the work of [3] and [5,6] for the loss, used the same algorithm (power iteration) as [3,4,1] to compute the Lipschitz, and proposed a new linear layer that seems to have good properties. They used several tricks (epsilon scheduler presented in Appendix B1, normalization trick presented in Appendix B2) without ablation study. With all this, they showed that it is possible to improve the certified accuracy for $\epsilon = 36/255$ and managed to achieve very good certified accuracy on ImageNet, which has not been done before. I find the results of this paper interesting because I think the overall problem is interesting and important, but I think the paper (in its current form) has little impact on the topic of certified accuracy with deterministic approaches. To improve the research value of the paper, the authors should provide a comprehensive ablation study and explain how and why these different techniques, when combined together, significantly improve certified accuracy.

[1] Meunier et al., A Dynamical System Perspective for Lipschitz Neural Networks ICML 2022
[2] Araujo et al., A Unified Algebraic Perspective on Lipschitz Neural Networks, ICLR 2023
[3] Tsuzuku et al., Lipschitz-Margin Training: Scalable Certification of Perturbation Invariance for Deep Neural Networks, NeurIPS 2018
[4] Farnia et al., Generalizable Adversarial Training via Spectral Normalization, ICLR 2019
[5] Leino et al., Globally-Robust Neural Networks, ICML 2021
[6] Singla et al. Improved deterministic l2 robustness on CIFAR-10 and CIFAR-100, ICLR 2022
[7] Trockman et al., Orthogonalizing Convolutional Layers with the Cayley Transform, ICLR 2021
[8] Singla et al, Skew Orthogonal Convolutions, ICML 2021
[9] Prach et al., Almost-Orthogonal Layers for Efficient General-Purpose Lipschitz Networks, ECCV 2022
[10] Huang et al., Training certifiably robust neural networks with efficient local Lipschitz bounds, NeurIPS 2021


**Questions:**

The authors have combined many different techniques and tricks to achieve their certified robustness. An ablation study would be interesting to identify those that increase the certified robustness.

- What is the performance of the author's architecture with an SLL layer that is 1 Lipschitz?
- What is the performance of the author's architecture with the loss from [3]? How does the EMMA loss improve the VRA?

The authors added some tricks for better training: the epsilon scheduler presented in Appendix B1 and the normalization trick presented in Appendix B2.
- How does the epsilon scheduler affect the final certified accuracy?
- How does the normalization trick affect the final certified accuracy?

- What is the certified accuracy for other perturbation thresholds? (e.g. $72/255$, $108/255$, $1$ to allow comparison with other work)

[3] Tsuzuku et al., Lipschitz-Margin Training: Scalable Certification of Perturbation Invariance for Deep Neural Networks, NeurIPS 2018

**Limitations:**

The authors discussed the limitation of robust classification.

---

> ### Author Rebuttal · Authors · 2023-08-10
>
> > On the efficiency of the approach
>
> Perhaps *efficient calculation* in the following sentence from the Abstract causes some degree of misunderstanding:
>
> “A key challenge in certifying ResNets is efficient calculation of the Lipschitz bound for residual blocks.”
>
> We would like to revise it as, “A key challenge in certifying ResNets is **efficiently calculating a _tight_ Lipschitz bound for residual blocks.**”
>
> We are not claiming that using the power method is our contribution. The power method is more efficient than orthogonalization, which is relatively expensive, and used by a lot of prior work. However, as we point out, power iteration is not sufficient to provide _tight_ bounds on _traditional_ residual blocks. Our primary contribution is to identify the source of the problems faced by residual blocks, and propose a new design for residual blocks that achieves the necessary properties to avoid these pitfalls. Taken together, our proposed LiResNet architecture can be seen as _enabling_ efficient and tight bound-calculating with the power method, as the same results cannot be achieved using the same efficient bounds without our architecture (see the comparison of ResNet to LiResNet).
>
> We will comment more on the computational cost in the author comment.
>
> > On the tightness of the bound
>
> It is correct to say PM computes the an **exact** Lipschitz bound for $x+Wx$ if one uses the reparameterization trick to directly compute $Lip(I+W)$ instead of using the upper-bound $1+Lip(W)$, which is often used in conventional ResNet blocks. Thanks for the suggestion and we will make this point clearer in writing.
>
> Additionally, we would like to share more results to showcase how switching from the conventional ResNet to LiResNet increases the tightness of the bounds:
>
> We compare the empirical lower-bound of the Lipschitz Constant, obtained by maximizing |f(x) - f(y)|/|x - y| w.r.t randomly initialized $x$ and $y$ till convergence for a network $f$. Here are results when $f$ is a LiResNet or a ResNet trained on CIFAR-10. “UB” (upper bound) is our Lipschitz constant estimation and “LB” is the lower bound from maximizing |f(x) - f(y)|/|x - y|.
>
> |          | LB    | UB    | LB / UB |
> |----------|-------|-------|---------------------|
> | LiResNet | 53.35 | 59.03 | 0.90                |
> | ResNet   | 23.93 | 101.4 | 0.24                |
>
> From this table, we see that LiResNet can have a much tighter Lipschitz constant estimation than conventional ResNet.
>
> > On the new EMMA loss
>
> EMMA could be considered simple, but our proposal is not the result of trivially combining existing works. LMT has existed for a long time but has not been shown to perform well. Few following studies have used this loss function. Our paper chronicles a progressive evolution of methodologies leading up to the SoTA VRA, transitioning from LMT to EMMA. We have discovered that by using the LC of the margin and adjusting epsilon dynamically, there are consistent and incremental improvements in certifiable robustness. By delving into the training dynamics and observing how the second highest classes—termed as the “threatening classes”—rotate throughout iterations, we've pinpointed a suboptimal aspect in the TRADES loss function. Furthermore, we illustrate why EMMA is especially advantageous for problems with a larger number of classes, where existing loss functions like TRADES can grapple with the phenomenon portrayed in Figure 2b. We hope that the intricacies and depth of our exploratory efforts in training certified robustness aren't overshadowed by the apparent simplicity of the technique we advocate for.
>
> > What is the performance of the author's architecture with an SLL layer that is 1 Lipschitz?
>
> We take the reviewer’s suggestion to replace linear residual blocks with SLL residual blocks and leave everything else the same (i.e. the whole network still has 12 convolution layers with 512 channels and 2 dense layers of 2048 neurons). We train both models in the same way (without any epsilon scheduler, DDPM, and etc.) and report the VRAs for a list of radii as suggested by the reviewer. For SLL models, we tried the models with 1) their default settings and 2) our optimization settings and reported the best VRAs. Here are our results on CIFAR-10/100 (other results will be in the revision of the paper):
>
> |           | epsilon | Using LiResNet Block | Using SLL Block  |
> |-----------|---------|----------------------|------------------|
> |           |  36/255 |   66.1               | 59.5             |
> | CIFAR-10  |  72/255 |   54.1               | 50.6             |
> |           | 108/255 |   45.2               | 42.2             |
> |           |  36/255 |   37.5               | 30.2             |
> | CIFAR-100 |  72/255 |   27.9               | 22.1             |
> |           | 108/255 |   22.1               | 16.5             |
>
>
>
> > How does the epsilon scheduler affect certified accuracy?
>
> Note that all of our ablation studies use the same epsilon scheduler setting. Thus, the difference between the VRAs of TRADES/ConvNet and EMMA/LiResNet in Table 2 reflects a difference that cannot be accounted for by the epsilon scheduler. On our largest model L12W512, turning the scheduler on introduces 0.4% VRA improvement on CIFAR-10.
>
> > How does the normalization trick affect certified accuracy?
>
> We follow the same setting from [1, 2], which is a widely used method for normalization-free methods. Please note that this method is also applied by SLL (the diagonal scaling matrix q in Equation 8 from SLL paper). Again, all of our ablation studies use this same setting. On our largest model L12W512, using the normalization-free methods can introduce 0.8% VRA improvement on CIFAR-10.
>
> [1] Shao et al. Is normalization indispensable for training deep neural network, NeurIPS 2020
>
> [2] Zhang et al. Fixup Initialization: Residual Learning Without Normalization. ICLR 2018

---

> > ### Comment · Reviewer_YimP · 2023-08-14
> > **Thank you for this extensive rebuttal. Some further comments.**
> >
> > Thank you for this extensive rebuttal. Here are some further comments:
> >
> > 1. **On the tightness of the bound** : thank you for providing this experiment, however, in my opinion, it is not even necessary to provide this comparison. The looseness of the bound on the residual layer with ReLU comes from the nonlinear activation, by removing it the calculation of the Lipschitz of the layer becomes exact. It is actually straightforward. Nevertheless, a small discussion must be provided in the paper: computing the exact bound using power iteration, the PM can approximate the Lipschitz of the layer with arbitrary precision with respect to the number of iterations. A small number of iterations can be done during training as the PM can converge progressively during training. A large number of iterations can be performed during inference as the values can be cached given they are independent of the input.
> > 2. **Comparison between SLL and LiResNet**: I strongly believe that this experiment was missing in the original version of the paper. It justifies why LiResNet is a good idea and useful. The remaining experiments are, in my opinion, further optimizations to improve the accuracy. I would suggest expanding the discussion on state-of-the-art ResNet-Like Lispchitz layers (which is almost non-existent in the current version) and explaining the difference between LiResNet and SLL.
> > 3. **Regarding the title of the paper**: The authors insist on the "depth" and on "unlocking certified accuracy on ImageNet". First, from Table 2 (b), the increase in certified accuracy with respect to the depth is minimal, so I would assume that certified accuracy clearly saturates very quickly. Furthermore, it is not very clear that depth allows for an increase in certified accuracy, it could also be the increase in the number of parameters. Therefore, I am not sure that the authors should focus on this specific parameter. Second, the title claims the first certified accuracy with ImageNet, which is incorrect. Randomized smoothing has provided probabilistic certificates on ImageNet for a long time. What the authors have shown is the first method that provides certified accuracy on ImageNet with _deterministic_ certificates and this is not emphasized enough. Maybe a better title would be: "Unlocking Deterministic Robustness Certification on ImageNet" or something similar.
> >
> > I will raise my score and encourage the authors to update the paper accordingly.

---

> > > ### Author Response · Authors · 2023-08-21
> > > **Thank you for your responses.**
> > >
> > > Thanks for providing us with more feedback! We would love to add the comparisons with SLL nets and the discussion on LiResNet v.s. SLL nets to the main body of the paper once we are giving a chance to update the writing. The discussion of PM will also go in the writing in the near feature. We thank for the suggestions on the title. "Unlocking Deterministic Robustness Certification on ImageNet" seems to be a good one. We will reword a bit but the reviewer's concern on the deterministic v.s. probabilistic aspect makes sense to us.
> > >
> > > Thank you!

---

### Official Review · Reviewer_jf6o · 2023-07-04

**Soundness:** 2 fair
**Presentation:** 3 good
**Contribution:** 3 good
**Rating:** 5
**Confidence:** 4

**Summary:**

This paper proposes Linear ResNet (LiResNet) and Efficient Margin Maximization (EMMA) loss for scalable training of provably robust deep neural network classifiers. With these two contributions, this work is able to achieve SOTA deterministic VRA on medium-to-large classification tasks.

**Strengths:**

- The paper is clearly written.
- The proposed method demonstrates strong empirical performance.

**Weaknesses:**

I am willing to raise the score by 1 or 2 points if the authors answer my questions satisfactorily.

**Weakness 1 : Ambiguity regarding the scalability of LiResNets.**
- The authors do not exactly pinpoint why LiResNets are scalable. In Table 2 (b), the authors show VRA  of ConvNets, ResNets, and LiResNets at various depths. Then, the authors just describe the table as-is, without any further discussion or analysis. Why do ConvNets diverge? Is it because of vanishing or exploding gradients? Why do ResNets have lower VRA than LiResNets? Is it because LiResNets enable tighter Lipschitz constant estimation? Is it because LiResNets allow faster convergence to optima? If there are multiple factors at work, how much does each factor contribute to the final VRA?

**Weakness 2 : Hand-wavy logic regarding the rotating threatening class phenomenon and EMMA.**
- Section 5.2, line 282, "rotating threatening class phenomenon observed during training may contribute to suboptimal training." --> Table 2 (a) only shows VRA before and after applying EMMA. How do we know whether performance gain comes because EMMA prevents the rotating threatening class phenomenon? Only a hand-wavy explanation in Section 4, line 209-215 is given. I would appreciate a more rigorous theoretical or experimental analysis of EMMA.

**Concern 1 : Representation power of LiResNets.**
- I get that LiResNet can admit tighter Lipschitz constant estimation. However, how scalable is LiResNet compared to ResNet in terms of clean accuracy when trained with cross entropy? Doesn't using linear residual blocks reduce the representation power of ResNets? If LiResNets have less representation power than ResNets, wouldn't we eventually have to return to ResNets when we have better training techniques and aim for even higher VRA?

**Questions:**

**Question 1** : Power method is used throughout training to estimate the global Lipschitz constant. How many iterations of power method is used at each step of training? How tight is the Lipschitz constant estimate?

**Question 2** : It is widely known that there is an accuracy-robustness trade-off. TRADES offers a hyper-parameter to control that trade-off. Would it be possible to control the trade-off for EMMA as well? How does the accuracy-robustness pareto frontier for LiResNet+EMMA compare to other methods?

**Limitations:**

Discussed in Section 6.

---

> ### Author Rebuttal · Authors · 2023-08-10
>
> > The authors do not exactly pinpoint why LiResNets are scalable ... Why do ConvNets diverge? Is it because of vanishing or exploding gradients?
>
> Yes, it is because of exploding gradients. Even for standard (non-robust) cross entropy training, ConvNets with 18 layers perform worse than ResNets. The task of certificate robustness is more difficult than standard cross entropy training since the training objective has a strong regularization from the network Lipschitz estimation, making the training of ConvNets even more difficult.
>
> > Why do ResNets have lower VRA than LiResNets? Is it because LiResNets enable tighter Lipschitz constant estimation? ... If there are multiple factors at work, how much does each factor contribute to the final VRA?
>
> There might be other factors behind why LiResNets enjoy higher VRA than ResNets, but the tighter Lipschitz bounds is definitely the dominating one. The Lipschitz constant lower bound for a conventional ResNet is much larger than the actual Lipschitz constant of the network, thus much of the model’s expressiveness is wasted. You can think of it as though a ResNet is trained to be robust at a much larger noise radius, but we can still only verify at a small noise radius. This also means ResNets will suffer greatly from overregularization during certified training. In our experiments, ResNets have converged when we report their VRAs so we suspect the convergence speed is not a major factor. More discussion on ResNet vs. LiResNet performance is included in our updated paragraph above.
>
> > Section 5.2, line 282, "rotating threatening ..." --> Table 2 (a) only shows VRA before and after applying EMMA. How do we know whether performance gain comes because EMMA prevents the rotating threatening class phenomenon? .... I would appreciate a more rigorous theoretical or experimental analysis of EMMA.
>
> Thanks for pointing out there might be missing steps in our reasoning line. By plotting the percentage of samples that the penalized non-label logits change in two consecutive epochs, Figure 2b locates the **rotating threatening class (RTC)** issue in the existing GloRo training with TRADES loss. **We run the same experiment with EMMA loss and plot the result .** Please check Figure 1 in the attached PDF file in the global rebuttal section.  We found that RTC happens to less training points if using EMMA loss, compared to TRADES loss. We hereby empirically validate that EMMA loss helps to reduce the times of having RTCs during training, mitigating the suboptimal issue raised in the paper. The mitigation of RTC is expected for GloRo Nets to converge smoothly and faster to achieve higher VRA, which is later evidenced by Table 2(a). Taking Figure 2b and Table 2 from the paper and the new plot (Figure 1 attached in the PDF) comparing EMMA loss and TRADES loss, we demonstrate an empirical correlation between the mitigation of RTC and the improvement of VRA. We will also clarify this reasoning line in the writing.
>
> > How scalable is LiResNet compared to ResNet in terms of clean accuracy when trained with cross entropy? Doesn't using linear residual blocks reduce the representation power of ResNets?
>
> It is not clear that a LiResNet is meaningfully less expressive than a traditional ResNet. Here we provide some results comparing LiResNet and ResNet on ImageNet classification, which show that LiResNet is capable of achieving similar performance of a ResNet and a VGG Net around the similar network depth.
>
> |                  | number of layers | Top 1 accuracy |
> |------------------|------------------|----------------|
> | LiResNet         |       18         |     73.3%      |
> | VGG              |       19         |     74.2%      |
> | ResNet           |       18         |     69.8%      |
>
> > How many iterations of the power method is used at each step of training? How tight is the Lipschitz constant estimate?
>
> We use 10 iterations during training. We empirically verified the tightness of our Lipschitz constant estimation of the entire network by optimizing |f(x) - f(y)|/|x - y| w.r.t x and y on the network f. Here are the results of our trained LiResNet models on 4 datasets, which show that the upper bound is fairly tight. “UB” (upper bound) is our Lipschitz constant estimation and “LB” is the lower bound from maximizing |f(x) - f(y)|/|x - y|.
>
> |    | ImageNet | CIFAR-10 | CIFAR-100 | Tiny-ImageNet |
> |----|----------|----------|-----------|---------------|
> | LB |   1.12   |  53.35   |   12.95   |     7.16      |
> | UB |   1.54   |  59.03   |   14.03   |     7.73      |
>
> > It is widely known that there is an accuracy-robustness trade-off. TRADES offers a hyper-parameter to control that trade-off. Would it be possible to control the trade-off for EMMA as well?
>
> Yes, we can manage this trade-off. For example, we can use a weighted sum of EMMA loss and cross entropy loss: CE-loss + k * EMMA-loss for some hyper-parameter k. To make our method simpler, we only present EMMA loss in the paper.
>
> > How does the accuracy-robustness pareto frontier for LiResNet+EMMA compare to other methods?
>
> It is hard to compare the pareto optimal of our method and other methods theoretically. However, we want to emphasize an important difference between our work and other works like CPL and SLL. While these methods enforce the Lipschitz Constant's regulation by imposing constraints on the weights, we opt to regulate it through the loss function, like in Leino et al. (2021). This method of regularizing Lipschitz constants with the loss has a potential advantage: it enables the learning of robust models with various Lipschitz constants. Should 1-Lipschitz Nets prove to be the optimal choice for certified robustness within certain data distributions, our models retain the ability to learn such functions. In our empirical evaluations from Table 1, our method is better than the SoTA (CPL and SLL) in both clean accuracy and VRA with a smaller model size.

---

> > ### Comment · Reviewer_jf6o · 2023-08-11
> > **Updated Score**
> >
> > The authors have answered my questions satisfactorily, and I have raised the score by a point.

---

> > > ### Author Response · Authors · 2023-08-11
> > >
> > > Thanks for reading our response and increasing the score.

---

### Official Review · Reviewer_diKy · 2023-07-06

**Soundness:** 3 good
**Presentation:** 3 good
**Contribution:** 2 fair
**Rating:** 4
**Confidence:** 3

**Summary:**

- The paper investigates Lipschitz-based Certification of neural networks.
- Authors aim to certify ResNets by extending the techniques from GLORO.
- Authors note that it is difficult to come up with a tight approximation of the residual block
- So authors replace the non-linear residual block to a linear block. They are then able to adapt GLORO results
- Authors add the non-linearities after the linear residual block.

**Strengths:**

1. Good writing
- The paper is well written. I could follow all the sections in one pass.
- Authors have now also added sections discussing transformers and complications with verifying them.

2. Thorough experiments
- A wide range of experiments are conducted on various datasets.
- The experimental results look decent.
- I have a question regarding the clean accuracy, which I have added in the questions section.

3. Good problem
- Certifying residual blocks is definitely worth doing.
- As authors mentioned, residual connections are also used in Transformers.

**Weaknesses:**

1. about the effectiveness of LiResNet
- Main trouble in verification comes from the non-linearity. It feels natural that removing non-linearities will make things easier. The authors are not really verifying the resnet as such.
- If $x + conv(x)$ can be written as a conv layer, then are you really verifying residual connections or just verifying a linear layer?


**Questions:**

- Regarding the clean accuracy, why is it higher than the baselines?
- Can you compare the numbers with networks of the same size?
- This will explain whether having such a network (as opposed to usual ResNet) is useful more broadly.



**Limitations:**

Yes, limitations are discussed well.

---

> ### Author Rebuttal · Authors · 2023-08-10
>
> We thank the reviewers for their valuable comments. Our answers to your questions are below.
>
> > Main trouble in verification comes from the non-linearity. It feels natural that removing non-linearities will make things easier. The authors are not really verifying the ResNet as such.
>
> We would like to highlight that main issue with the Lipschitz-based approaches on conventional ResNet is not originated from the **nonlinearity** of the module – it comes from bounding the addition of the residual branch and the skip connection $y = x + f(x)$, which is a linear part in a ResNet block. The Lipschitz upper-bound of such a linear combination, $Lip(f+g)\leq Lip(f) + Lip(g)$, is shown to be loose in the paper and motivates our design of linear residual block. The characterization that we are “not verifying ResNet” is inaccurate, given that LiResNets consist of the classic skip connections that enable gradients to flow back to the deeper part of the model without vanishing issues, which is the defining feature of a “ResNet” architecture; though of course, the LiResNet architecture is a specific innovation that achieves desirable properties not shared with a traditional ResNet architecture.
>
> > If  x + conv(x) can be written as a conv layer, then are you really verifying residual connections or just verifying a linear layer?
>
> We are verifying the entire network, not just a linear layer. Moreover, because the LiResNet blocks are the equivalent of x + conv(x), **these layers do contain residual connections, hence why the LiResNet is a residual network. It is merely a reparameterization technique to write these layers as a convolution**; however, _this does not justify saying that this makes it just a convolutional network_, as the reparameterization acts as a constraint on which convolutions will be learned. This is exactly analogous to how a convolutional layer can be written as a dense layer, while clearly a convolutional network is not the same as a dense network in any meaningful way. Our ablation study (LiResNet vs. ConvNet) shows that this reparameterization indeed has better training properties than directly training a ConvNet.
>
> > Regarding the clean accuracy, why is it higher than the baselines?
>
> We would appreciate it if the reviewer specifies which result (e.g. in which table) is referred to in the question. If the question is pretty general, we argue that it is that LiResNets facilitate a better trade-off between expressiveness (depth) and tight Lipschitz bounds, allowing us to learn a robust function with less overregularization; thus the clean accuracy is also higher.
>
>
> > Can you compare the numbers with networks of the same size?
>
> In our ablation studies, we do compare the numbers with networks of the same size. When compared with previous work, it is hard to compare with exactly the same size since the backbone is different. In Table 1, we compare our results with the best reported results from previous work. Specifically, to compare with SLL, the current state of the art result, our method achieves better VRA with a much smaller model size (in terms of parameters), which shows the effectiveness of our method. See the following table for comparing VRAs under eps=36/255 between ours and SLL summarized from our paper.
>
> |  Models           | Model Size | CIFAR-10 | CIFAR-100 | Tiny-Imagenet |
> |-------------|------------|----------|-----------|---------------|
> | LiResNet L12W512    | 49M        | 70.1     | 41.5      | 33.6          |
> | SLL Small   | 41M        | 62.6     | 34.7      | 19.5          |
> | SLL X-Large | 263M       | 65.8     | 36.5      | 23.2          |
>
> Here is a comparison of training speed. Our largest model and the smallest SLL model (SLL Small) have a close training throughput while the largest SLL model is 4.5 times slower than our LiResNet.
>
> |                                | SLL Small | SLL X-Large | LiResNet L12W512 |
> |--------------------------------|-------|-------------|----------|
> | Training speed (images/second) |  3943 |     813     |   3805   |

---

> > ### Comment · Reviewer_diKy · 2023-08-21
> > **Comment**
> >
> > Hi,
> >
> > Thanks to the authors for their response.
> >
> > 'We would like to highlight that main issue with the Lipschitz-based approaches on conventional ResNet is not originated from the nonlinearity of the module – it comes from bounding the addition of the residual branch and the skip connection
> > , which is a linear part in a ResNet block.'
> > If the problem is not coming from the non-linearity, then why are you able to get a tighter bound when you replace g(x) with conv(x)?  For r(x) = x + g(x), as long as g is a linear layer of a conv layer, you can get a tight bound. As soon as g(x) contains a non-linearity, your bound is loose. So you replace it with a simple linear/conv layer. Is this not what you do?
> >
> > 'This is exactly analogous to how a convolutional layer can be written as a dense layer, while clearly a convolutional network is not the same as a dense network in any meaningful way.' Yes, it can. In verification papers, for deriving bounds for conv layers, they are indeed treated as linear layers. but there the non-linearity comes before the residual connection not after. If you take r(x) = x + g(x), where g(x) is a conv and non-linearity, then this cannot be written as a linear layer.
> >
> > Regarding the clean accuracy, why is it higher than the baselines?
> > This is an important question. Firstly, I would like to decouple the architecture from the method.
> > Secondly, I don't quite understand why the network has better accuracy than normal ResNet. Is the claim that your network is better than ResNet? That is a very strong claim. I barely understant what this means ' facilitate a better trade-off between expressiveness (depth) and tight Lipschitz bounds, allowing us to learn a robust function with less overregularization; thus the clean accuracy is also higher.'
> >
> > Can you compare the numbers with networks of the same size?
> > I would like to understand in more detail, what is causing the difference in results. This relates to my previous question. I have already seen the results from the paper.

---

> > > ### Author Response · Authors · 2023-08-22
> > > **Response to the follow-up questions (1/2)**
> > >
> > > _High-level Comment_
> > >
> > > Based on the discussion, we believe there is confusion regarding the primary contribution of this work. We are not proposing a new certification technique. Rather, we
> > > - (1) identify a fundamental problem for certified training of ResNets, and
> > >
> > > - (2) propose a new type of residual architecture that solves this problem. The LiResNet architecture is specifically made to make existing certification techniques (e.g., GloRo) more effective and scalable, and we show that it can lead to significant increases in the SoTA for deterministically certified accuracy.
> > >
> > > _Responses to Questions_
> > >
> > > > If the problem is not coming from the non-linearity, then why are you able to get a tighter bound when you replace g(x) with conv(x)? For r(x) = x + g(x), as long as g is a linear layer of a conv layer, you can get a tight bound. As soon as g(x) contains a non-linearity, your bound is loose. So you replace it with a simple linear/conv layer. Is this not what you do?
> > >
> > > Perhaps we misunderstood your original question. It is true that we remove the nonlinearity from inside the residual block (making the residual branch linear) and place it after the skip connection is joined. But this does *not* remove the nonlinearities from the network as a whole, so they can still introduce looseness, but not in a way that is as problematic as when the bound is estimated as the sum of the residual and skip branches. We suspect the major concern here is about the term “ResNet.” It is true that we are not verifying the “conventional” ResNet as proposed in [1]; however, our LiResNet block maintains the skip connections, which is why this is a true residual architecture. E.g., according to the original authors in [1]: “Formally, in this paper we consider a building block defined as: $y = F(x, \{W\}) + x$." Notably there is no stipulation that $F$ must be nonlinear. The most conventional ResNet, “[in] the example in Fig. 2..., $F = W_2\sigma(W_1 x)$" is clearly simply an instantiation of the general residual framework. Regardless, the goal in this work is not to certify a specific architecture, but rather, to make the benefits of residual networks possible in the context of certified training in order to train deeper certified networks with higher VRA. We will try to clarify the writing so there is not confusion over what we mean by “verifying ResNet.”
> > >
> > > > 'This is exactly analogous to how a convolutional layer can be written as a dense layer, while clearly a convolutional network is not the same as a dense network in any meaningful way.' Yes, it can. In verification papers, for deriving bounds for conv layers, they are indeed treated as linear layers. but there the non-linearity comes before the residual connection not after. If you take r(x) = x + g(x), where g(x) is a conv and non-linearity, then this cannot be written as a linear layer.
> > >
> > > We believe you might have misunderstood our reply. Any conv layer can be written as a dense layer. But this doesn’t mean there is no meaningful difference between a conv layer and a dense layer. This is just as true for a LiResNet block. It can be written as a conv (or dense) layer by reparameterization, but it adds an additional constraint that allows the gradient to flow through the skip connection, thus it is meaningfully different from a conv layer.
> > >
> > > By saying there is a meaningful difference, we don’t mean it requires a new certification technique. Of course, LiResNet blocks can have their LC bounded in the same way as other linear layers (e.g., convolutions), but note that the novel part of our work is not the use of the power method for obtaining the bound on the LC of a layer, which has been used by prior work.

---

> > > > ### Author Response · Authors · 2023-08-22
> > > > **Responses to the follow-up questions (2/2)**
> > > >
> > > > > Regarding the clean accuracy, why is it higher than the baselines? This is an important question. Firstly, I would like to decouple the architecture from the method. Secondly, I don't quite understand why the network has better accuracy than normal ResNet. Is the claim that your network is better than ResNet? That is a very strong claim. I barely understant what this means ' facilitate a better trade-off between expressiveness (depth) and tight Lipschitz bounds, allowing us to learn a robust function with less overregularization; thus the clean accuracy is also higher.'
> > > >
> > > > We are still unclear what comparisons you are asking for (also, we would like to point out that the architecture is the major contribution, not the way the network is certified—we did not propose a new certification method). Our results do indicate that LiResNet is better than standard ResNet **for certified training**. This is because the LC estimation is too loose on the standard ResNet, leading to over-regularization and loose certificates; the former negatively impacts the clean accuracy, and both negatively impact the VRA. LiResNet allows us to train deep networks while maintaining a tighter bound, which leads to less unnecessary regularization, and tighter certificates.
> > > >
> > > > We did not claim that LiResNet is better than ResNet in settings where we are not training specifically for certified robustness. Note that the models in our evaluation are trained for certified robustness, and “clean accuracy” is measured on these same models, not on models that were trained using standard (non-robust) training (this is standard practice in the literature on deterministic robustness certification).
> > > >
> > > > Regarding the comparison to the “baseline” we are unsure what baseline you are referring to, and which models you believe don’t have a comparison to a similarly-sized model. *As it approaches the deadline of the rebuttal phase and in case we are not able to continue the conversation, we have provided a few alternative replies below*:
> > > >
> > > > - **If by “the baseline” you mean the ResNet column in table 2b**, the answer is that these ResNet and LiResNet architectures are roughly the same size, and the difference in performance primarily stems from the looser bound on ResNets. As mentioned, the looser bound leads to over-regularization: when the bound is loose, the model effectively has to certify at a much larger radius. Because we empirically observe a robustness-accuracy trade-off (as referenced by reviewer jf6o), certifying at a higher effective radius leads to the model sacrificing clean accuracy.
> > > >
> > > > - **If by “the baseline” you mean the ResNet rows in table 1**, those rows use a different approach to certifying standard ResNets from prior work (SLL). In table 3 in the attached pdf (in the global Rebuttal), we provide an apples-to-apples comparison of our architecture with the LiResNet blocks replaced by SLL layers (which yields a more conventional ResNet). Here we see that the performance of the conventional ResNet (SLL) is a lot lower than the LiResNet.
> > > >
> > > > Overall, we hope our responses are helpful to address the reviewer's concerns in the contributions and some empirical results. We are grateful for the reviewer's feedback on the paper and our conversations should be reflected in the writing when we are given a chance to do so in the near future.
> > > >
> > > > #### Reference
> > > > [1] Kaiming He et al. Deep Residual Learning for Image Recognition, CVPR 2015.

---

### Official Review · Reviewer_xiEp · 2023-07-07

**Soundness:** 3 good
**Presentation:** 4 excellent
**Contribution:** 3 good
**Rating:** 7
**Confidence:** 4

**Summary:**

This paper aims to scale up deterministic certified robustness to deeper neural networks (ResNet) and more complicated datasets (ImageNet). To this end, the authors proposed a new residual block named LiResNet and a new loss function named Efficient Margin Maximization. The proposed method achieves the state of the art results on various benchmarks.

**Strengths:**

This paper is very well written and easy to follow. Each design choice is well motivated and thorough experiments demonstrated the effectiveness of the proposed methods.
- The proposed LiResNet architecture is a simple but effective technique to bypass the difficulty in certifying ResNet. Experiments also showed its scalability to deeper networks.
- The new loss is well motivated by the experiment finding that the inconsistency of the threatening classes increases when number of classes increases.
- Comprehensive experiments on various datasets have shown the effectiveness of the proposed methods and established new state-of-the-art.

**Weaknesses:**

- Despite the effectiveness of LiResNet in certified robustness, it may be still less expressive and scalable than the actual ResNet. It would be good to discuss the limitation of the LiResNet architecture: what do we sacrifice by "linearizing" the skipping connection?
- The new proposed loss is well motivated, but it seems that in the experiment there is no ablation study on it. It'd be good to include Gloro + LiResNet but without EMMA to show the improvement of EMMA compared to the Gloro loss.

**Questions:**

Is there any normalization layer in the LiResNet architecture? There seems to be none in Figure 2(a). It is a bit surprising the it can scale so well in depth without any normalization layers.

**Limitations:**

Yes.

---

> ### Author Rebuttal · Authors · 2023-08-10
>
> We thank the reviewers for their valuable comments. Our answers to your questions are below.
>
> > Despite the effectiveness of LiResNet in certified robustness, it may be still less expressive and scalable than the actual ResNet. It would be good to discuss the limitation of the LiResNet architecture: what do we sacrifice by "linearizing" the skipping connection?
>
> It still remains an open question if a LiResNet is meaningfully less expressive than a conventional ResNet. Here we provide some results comparing LiResNet and ResNet on ImageNet classification, which show that LiResNet is capable of achieving similar performance of a ResNet and a VGG-Net around the similar network depth. We follow the standard training settings to train classification models on ImageNet.
>
>
> |                  | number of layers | Top 1 accuracy |
> |------------------|------------------|----------------|
> | LiResNet         |       18         |     73.3%      |
> | VGG [1]              |       19         |     74.2%      |
> | ResNet [2]          |       18         |     69.8%      |
>
> Perhaps a good way of thinking about this is the following: suppose we fix the number of convolutions in a network, but vary how many are placed inside each residual block. At one extreme, all the convolutions are in a single block, so we essentially have a ConvNet. On the other end, we have a LiResNet, with only one convolution per block (a traditional ResNet would usually have two or three). While the additional skip connections can be seen as additional constraints on the function learned by the model, it seems clear most of the expressiveness comes from the depth of the backbone regardless of the block size. In this view, the ConvNet is the most expressive, but we rarely worry that a ResNet has insufficient capacity by comparison, because the skip connections ultimately allow us to train a deeper network that wouldn’t be possible with a ConvNet.
>
>
> > The new proposed loss is well motivated, but it seems that in the experiment there is no ablation study on it. It'd be good to include Gloro + LiResNet but without EMMA to show the improvement of EMMA compared to the Gloro loss.
>
> We include an ablation study for the loss in Table 2 (a), where we compare EMMA and the TRADES loss (proposed by Leino et al [3] as the default loss in their work). The results are most notable on the tasks with more classes, as hypothesized.
>
> > Is there any normalization layer in the LiResNet architecture? There seems to be none in Figure 2(a). It is a bit surprising that it can scale so well in depth without any normalization layers.
>
> Yes, as perceived by the reviewer LiResNets is a normalization-free architecture. In fact, to the best of our knowledge, several works [3, 4, 5] in training certifiable robust models end up with not using any normalization layers. There are a few reasons. Firstly, one of the widely-used layers, layer normalization, is not Lipschitz-continuous so it does not fit into Lipschitz-based methods. Secondly, there are diminishing returns in using batch normalization from our experiments and from other papers [3, 4]. The following paragraph provides some explanations for the choice of normalization-free.
>
> While the reason for this is not 100% clear to the community, there is a large body of work on normalization-free methods, and some work suggests that gradient norm preservation (GNP) may play a similar role as compared to normalizations. GNP has been found as a fundamentally important building block for certifying networks and can be realized by using MinMax or GroupSort activations [6]. It may help to stabilize the internal activations by stabilizing gradients. Besides GNP, there is another line of work studying parameter initializations in order to realize normalization-free [7, 8]. Their insights result in some specific ways of initializing the residual block detailed in the Appendix B2.
>
>
>
> [1] Simonyan et al., Very Deep Convolutional Networks for Large-Scale Image Recognition, ICLR 2015.
>
> [2] He et al., Deep Residual Learning for Image Recognition, CVPR 2016.
>
> [3] Leino et al., Globally-Robust Neural Networks, ICML 2021.
>
> [4] Trockman et al., Orthogonalizing Convolutional Layers with the Cayley Transform, ICLR 2021.
>
> [5] Huang et al., Training certifiably robust neural networks with efficient local Lipschitz bounds, NeurIPS 2021.
>
> [6] Anil et al. Sorting out Lipschitz function approximation, ICML 2019.
>
> [7] Shao et al. Is normalization indispensable for training deep neural network, NeurIPS 2020.
>
> [8] Zhang et al. Fixup Initialization: Residual Learning Without Normalization. ICLR 2018.

---

> > ### Comment · Reviewer_xiEp · 2023-08-21
> >
> > Thanks the authors for their response. Although LiResNet is not as expressive as ResNet, it is a step towards scaling up certified robustness to larger architecture and more complicated dataset. Therefore, I lean towards accepting the paper.

---

### Author Rebuttal · Authors · 2023-08-10

We thank all reviewers for their valuable reviews. Here we respond to some questions that more than one reviewer are interested.

- Expressiveness of LiResNet in standard training

 We train LiResNet on ImageNet in the standard cross entropy setting and find that LiResNet is capable of achieving similar performance of a ResNet and a VGG-Net around the similar network depth. We follow the standard training settings to train classification models on ImageNet. Table 1 in the attached PDF shows the comparison with ResNet 18 and VGG 19.

- Comparison with prior work of the same size

When compared with previous work, it is hard to compare with exactly the same size since the backbone is different. In Table 1, we compare our results with the best reported results from previous work. Table 2 in the attached PDF shows the comparison of LiResNet and SLL (which is the current state of the art). SLL small has the similar model size and training speed with our largest LiResNet L12W512 while SLL X-Large is 4 times bigger and 4.5 times slower than our model. Our method can still outperform SLL X-Large under eps=36/255. The training speed is obtained using the official codes from SLL and the exact setting from the SLL paper.  All experiments are conducted on the same 4-GPU machine.

We further compare with SLL in a more fair setting (thanks Reviewer YimP for the motivation). We simply replace our proposed LiResNet block with SLL layer and use the same backbone setting: 12 convolutions with 512 channels and 2 linear layers with 2048 dimensions. In this comparison, our method and SLL has the same model size. Table 3 shows the comparison on CIFAR10 and CIFAR100 under different epsilons, and our proposed LiResNet block performs constantly better.  In this comparison, we do not use epsilon scheduling as Reviewer YimP wondered. For SLL models, we tried the models with 1) their default settings and 2) our optimization settings and reported the best VRAs.

---

### Decision · Program_Chairs · 2023-09-21

**Decision:**

Accept (poster)

**Comment:**

All reviewers agree that the studied problem is important and the experiments are thorough, though some reviewers have concerns on the expressiveness of LiResNets. The paper deserves acceptance. However, please revise the paper accordingly and address the reviewers' concerns in the camera-ready version.